# Earthquake-induced debris flows at Popocatépetl Volcano, Mexico

Velio Coviello[1,2], Lucia Capra[3], Gianluca Norini[4], Norma Dávila[5], Dolors Ferres[6], Víctor Hugo Márquez-Ramirez[3], Eduard Pico[3]

[1] Free University of Bozen-Bolzano, Facoltà di Scienze e Tecnologie, Bolzano, Italy
5 [2] now at Research Institute for Geo-Hydrological Protection, Consiglio Nazionale delle Ricerche, Padova, Italy
[3] Centro de Geociencias, Universidad Nacional Autónoma de México, Campus Juriquilla, Querétaro, México
[4] Istituto di Geologia Ambientale e Geoingegneria, Consiglio Nazionale delle Ricerche, Milano, Italy
[5] Escuela Nacional de Estudios Superiores, Universidad Nacional Autónoma de México, Campus Juriquilla, Querétaro, México
10 [6] Escuela Nacional de Ciencias de la Tierra, Universidad Nacional Autónoma de México, Ciudad de México, México

*Correspondence to*: Velio Coviello (velio.coviello@unibz.it)

**Abstract.** The 2017 $M_W$ 7.1 Puebla-Morelos intraslab earthquake (depth = 57 km) severely hit Popocatépetl volcano, located ~70 km North from the epicenter. The seismic shaking triggered shallow landslides on the volcanic edifice, mobilizing slope material saturated by the three-day antecedent rainfall. We produced a landslide map based on a semi-automatic classification of a 50-cm resolution optical image acquired two months after the earthquake. We identified hundreds of soil slips and three large debris flows for a total affected area of 3.8 km$^2$. Landslide distribution appears controlled by the joint effect of slope material properties and topographic amplification. In most cases, the sliding surfaces correspond with discontinuities between pumice-fall and massive ash-fall deposits from late Holocene eruptions. The largest landslides occurred on the slopes of aligned ENE-WSW-trending ravines, on opposite sides of the volcano, roughly parallel to the regional maximum horizontal stress and to volcano-tectonic structural features. This suggests transient reactivation of local faults and extensional fractures as one of the mechanisms that has weakened the volcanic edifice and promoted the largest slope failures. The material involved in the larger landslides transformed into three large debris flows due to liquefaction. These debris flows mobilized a total volume of about $10^6$ cubic meters of material also including large wood, were highly viscous, and propagated up to 7.7 km from the initiation areas. We reconstructed this mass wasting cascade by means of field evidence, samples from both landslide scarps and deposits, and analysis of remotely sensed and rainfall data. Although subduction-related earthquakes are known to produce a smaller number of landslides than shallow crustal earthquakes, the processes here described show how an unusual intraslab earthquake can produce an exceptional impact on an active volcano. This scenario, not related to the magmatic activity of the volcano, should be considered in multi-hazard risk assessment at Popocatépetl and other active volcanoes located along volcanic arcs.

## 1 Introduction

Earthquakes can induce large slope instabilities in tectonically active regions, resulting in a relevant source of hazard and damage. Earthquake magnitude (M) and the resulting intensity of ground vibration control the extent of the area where landslides may occur. One of the first comprehensive historical analysis of earthquake-induced landslides was done by Keefer (1984), who showed that the maximum area likely to be affected by landslides during a seismic event increases with M following a power law scaling relationship. In the following years, a growing number of studies started focusing on the impact of landsliding caused by large-magnitude earthquakes along relatively shallow crustal faults. In particular, it was observed that the fault rupture mechanism strongly influences the distribution of landslides, which usually are more abundant on the hanging-wall in case of reverse or normal faults (Sato et al., 2007) and present a symmetric distribution in case of strike-slip faulting (Xu and Xu, 2014). In the case of coseismic landslides related to earthquakes in subduction-zones, very few data and inventories are available (LaHusen et al., 2020; Schulz et al., 2012; Serey et al., 2019; Wartman et al., 2013). Best examples are the landslides induced by the 2010 Chile megathrust earthquake (Serey et al., 2019) and by the 2011 Tohoku Earthquake, Japan (Wartman et al., 2013). In both cases, thousands of shallow landslides were identified, but the main conclusion of these works is that the number of landslides generated by megathrust earthquakes is smaller than the number of events triggered by shallow crustal earthquakes by at least one or two orders of magnitude. Today, we know that the spatial distribution of earthquake-induced landslide is also a function of geological parameters (e.g., contrast in rock coherence, permeability), topography (slope and shape), land-cover and land-use, and ground-motion characteristics such as amplification and shaking frequency (Fan et al., 2019; Von Specht et al., 2019). Concerning the impact of earthquake on sediment-related hazards, a dramatic increase of sediment yield has been documented after large earthquake-induced landslides (Pearce and Watson, 1986; Dadson et al., 2004; Marc et al., 2019). Progressively, source areas on highlands can become quickly stable as fine material is removed and new vegetation grows and stabilizes the slope (Domènech et al., 2019) but debris flows can still occur due to remobilization of deposited material along the channel (Fan et al., 2021).

On active volcanoes, a large variety of factors can promote slope instability and failure such as magma intrusions, hydrothermal activity, gravitational spreading of the basements, climate fluctuations and regional tectonics (Capra et al., 2013; Mcguire, 1996; Norini et al., 2008; Roberti et al., 2017; Roverato et al., 2015). In particular, earthquakes are recognized to play one of the most important roles in the initiation of slope failures on volcanoes (Kameda et al., 2019; Sassa, 2005; Siebert, 2002). Volcanic slopes that are close to a critical state can be particularly susceptible to perturbations produced by regional earthquakes. Volcanic landslides include a wide spectrum of instability phenomena, from small slope failures to large sector collapse evolving into catastrophic debris avalanches. Intermediate processes such as shallow landslides and debris flows are common in the case of an earthquake, but they are relatively poorly documented for past events. Debris flows, often called lahar in volcanic environments, are usually associated with eruptions that induce ice/snow-melt or with intense rainfalls occurring during intra-eruptive phases (e.g., Capra et al., 2018; Major et al., 2016; Manville et al., 2009). Few examples of long-runout debris flows triggered by earthquakes have been described on active volcanoes

(Schuster et al., 1996; Scott et al., 2001). In Mexico, a M6.5 earthquake that occurred in 1920 induced several landslides in the Pico de Orizaba - Cofre de Perote volcanic chain that transformed into debris flows with catastrophic effects for villages along the Huizilapan ravine (Camacho, 1920; Flores, 1922). More recently, several thousands of shallow landslides were triggered by the Tecomán earthquake of 21 January 2003 (M 7.6) in the volcanic highlands north and northwest of Colima City (Keefer et al., 2006).

In this paper, we investigate the exceptional mass wasting episode triggered by the 19 September 2017, $M_W$ 7.1 Puebla-Morelos intraslab earthquake along the eastern and western sides of Popocatépetl volcano. The seismic shaking mobilized pre-existing ash and pumice fall deposits producing hundreds of coseismic soil slips. The largest ones had a total volume of about $10^6$ cubic meters and transformed into debris flows that traveled up to 7.7 km on the Western side of the volcanic edifice. This phenomenon, never studied before at Popocatépetl volcano, and probably unique on an active stratovolcano along a continental volcanic arc, has important implications for hazard assessment, as the actual hazard map only includes the impact of lahars related to volcanic activity (Martin Del Pozzo et al., 2017). In the following, we provide a general introduction to the geomorphology of Popocatépetl volcano and to its recent volcanic activity. Then, we describe the impact of the $M_W$ 7.1 Puebla-Morelos earthquake on the volcano slopes in terms of ground vibrations and landslide activity. Finally, we reconstruct the transformation of the major landslides into long runout debris flows and we discuss the hazards implication for an active volcano.

## 2 Background

### 2.1 Popocatéptl volcano

Popocatépetl volcano (19°03'N, 98°35'W; elevation 5,450 m a.s.l.) is located in the central sector of the Trans-Mexican Volcanic Belt (TMVB) and represents the active and southernmost stratovolcano belonging to the Sierra Nevada volcanic chain (Pasquaré et al., 1987) (Figure 1a, b). The Popocatépetl is a composite volcano and its present shape is the result of eruptive activity that rebuilt the modern cone after the 23.5 ka flank collapse (Siebe et al., 2017). During the Last Glacial Maximum (20-14 ka) glacier activity resulted in extensive moraines and glacial cirques (Vázquez-Selem and Heine, 2011). The lower part of the cone features a gentle slope (10-15°) and a dense vegetation cover up to approximately 3800 m a.s.l. (Figure 2a), where pine trees became scattered and surrounded by dense tropical alpine grasslands (*zacatonal alpino*, Almeida et al., 1994), that can measure up to 1 m in height. Then, the cone becomes progressively steeper (20-30°) and unvegetated up to the summit. In the upper portion of the cone, the slopes are covered by abundant unconsolidated ash named "Los Arenales" from the recent vulcanian eruptions (Figure 2a).

Quaternary volcanic activity of Popocatépetl volcano has been characterized by catastrophic episodes including sector collapses and Plinian eruptions that emplaced pyroclastic density currents and thick pumice fall deposits, predominantly toward the east and northeast (Figure 1b) (Siebe and Macías, 2006). Based on its Holocene eruptive record, Plinian eruptions at Popocatépetl have occurred with variable recurrence time of about 1–3 ka (Siebe et al., 1996). Since 1994, the volcano

entered in a new eruptive phase, which includes dome growths that are subsequently destroyed during strong vulcanian eruptions with columns up to 8 km in height, accompanied with ash fall that has been affecting populations in a radius of approximately 100 km. Eruptive activity played the primary role in accelerating the glacier retreat on the northern slope of the volcano (Julio-Miranda et al., 2008).

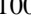

**Figure 1:** (a) Plate tectonic settings of Central Mexico (CP Cocos Plate, NAP North America Plate, RP Rivera Plate, PP Pacific Plate, MAT Middle American Trench) and location of Popocatépetl volcano (PV) in the Trans-Mexican Volcanic Belt (TMVB). (b) Details of the area affected by the $M_W$ 7.1 Puebla-Morelos earthquake and location of the seismic station CU and PPIG and of the rain gauge ALTZ. Popocatépetl volcano (PV) is the southernmost edifice of the Sierra Nevada volcanic chain along with the Iztaccihuatl (IzV), Telapón (Te) and Tláloc (Tl) volcanoes. The main distribution areas of the pumice fall deposits from the last two Plinian eruptions of the PV are indicated with white dashed lines (background image Landsat/Copernicus from Google Earth: ©Google 2020, ©INEGI 2020). (c) Strong motion recorded at station PPIG during the Puebla-Morelos earthquake on 19 September 2017 (channel HLZ refers to the vertical component, channel HLE and HLN to horizontal components).

In recent time, only two large lahar events were observed along the Huiloac Gorge (Hg, Figure 1b), in 1997 and 2001, associated with eruptive phases (Capra et al., 2004). At those times, the Ventorillo glacier was still present on the northern face of the volcano. Both lahars propagated to the town of Santiago Xalitzintla (SX, Figure 1b), located ~15 km E from the volcano summit. The 1997 lahar originated after a prolonged explosive activity with emission of ash, which caused the partial melt of the glacier. The rapid release of water gradually eroded the river bed and triggered a debris flow. The 2001

lahar originated from the remobilization of a pumice flow deposit emplaced over the Ventorillo glacier on the northern side of the volcano. The event occurred ~5 hrs after the pyroclastic flow emplacement, and the debris flow was characterized by a stable sediment concentration of 0.75 (Capra et al., 2004). In the distal part, the 1997 lahar transformed into a hyperconcentrated flow, while the 2001 one maintained the characteristics of a debris flow due to its apparent cohesion conferred by a silty-rich matrix inherited from the pumice flow deposit. Apart from the Huiloac Gorge, which was characterized by significant geomorphic transformations due to these latter processes (Tanarro et al., 2010), most of the drainage network of Popocatépetl volcano has a dense vegetation cover and presents stable, low-energy sediment transport conditions (Castillo et al., 2015). These stable conditions suddenly changed during the $M_W$ 7.1 Puebla-Morelos earthquake.

## 2.2 The MW 7.1 intraslab Puebla-Morelos earthquake

On 19 September 2017, central Mexico was hit by a $M_W$ 7.1 intraslab seismic event (depth = 57 km) named the Puebla-Morelos earthquake (Melgar et al., 2018; Singh et al., 2018). The epicenter of the earthquake was located ~70 km South from the summit of Popocatépetl volcano and ~100 km South from Mexico City (Figure 1b). The focal mechanism corresponds to a normal fault with a dip angle of 44°-47° (Melgar et al., 2018). The 2017 $M_W$ 7.1 Puebla-Morelos earthquake produced the most intense ground shaking ever recorded in Mexico City during a subduction-related earthquake, and was the most damaging event for this densely urbanized part of the country since the 1985 $M_W$ 8.1 Michoacán interplate earthquake, that occurred exactly 32 years before (Singh et al., 2018). The damage was surprisingly large in the critical frequency range for Mexico City (0.4–1 Hz), where the earthquake severely damaged hundreds of buildings and killed 369 people (Singh et al., 2018). The 2017 intraslab earthquake occurred closer to Mexico City, at greater depth, and involved higher stress drop than their interplate counterparts, such as the 1985 Michoacán event. The stress drops of intraslab events have been estimated as ~4 times greater than that of the interplate earthquakes (García et al., 2005) and the ground acceleration of the intraslab earthquakes are expected to be more enriched at higher frequencies than those of the interplate events (Furumura and Singh, 2002; Singh et al., 2018). During the 2017 $M_W$ 7.1 Puebla-Morelos earthquake, the Peak Ground Acceleration (PGA) recorded at station Ciudad Universitaria (CU) was the highest recorded in the last 54 years of observations (57.1 cm/s$^2$) (Singh et al., 2018). Station CU is located on the external boundary of the sedimentary basin responsible for the well-known seismic amplification at Mexico City (Figure 1b). The strong ground motion recorded at PPIG station (Figure 1c), located at 3980 m a.s.l. on Popocatépetl volcano slopes, featured a much higher value of PGA (106.83 cm/s$^2$, 0.1$g$) than the one observed at station CU.

## 3 Data and methods

We adopted a combined field- and remote-based approach to retrieve information about the earthquake impact on such a difficult to access environment. Semi-automated satellite-image classification is a rapidly developing tool producing reliable landslide maps (e.g., Fan et al., 2019 and references therein). We used optical satellite data to identify the main areas

affected by landslides and to constrain the timing of the landslide occurrences with respect to the earthquake event (Figure 2a, b). We constructed a preliminary landslide map (Figure 2c) based on the interpretation of an archive Pléiades 1A image (incidence angle of 14.63°, resolution of 0.5 m) acquired two months after the earthquake. A Normalized Difference Vegetation Index (NDVI) was calculated using band 1 (red) and band 4 (infrared). The resulting raster was classified for excluding vegetation cover, roads and buildings from the analysis and selecting only landslide scars or depositional areas. The final map (Figure 2c) was validated and refined based on data that we collected in the field. Most landslides are located on the W side, on the E side, and on the SE side of the volcanic edifice (Figure 3).

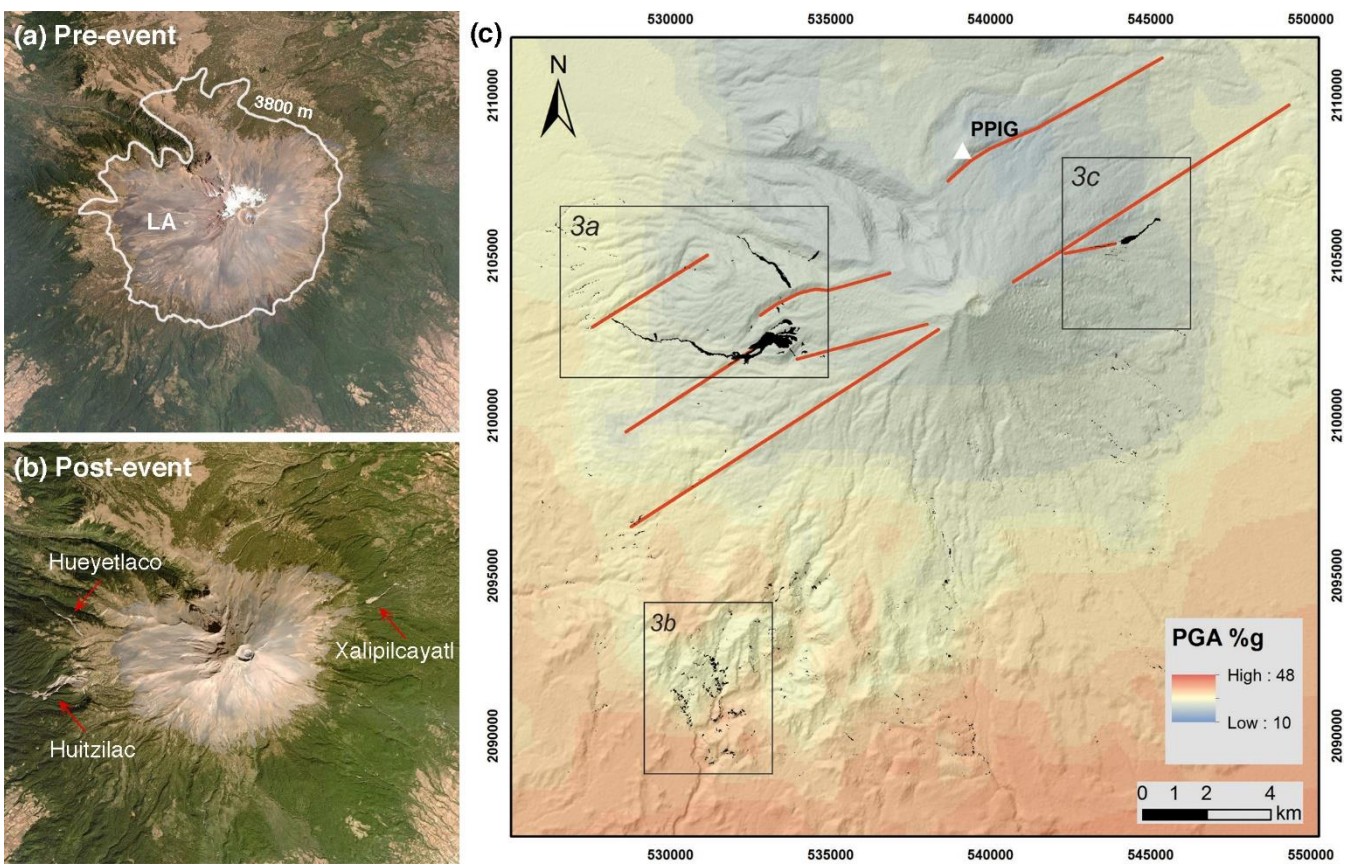

**Figure 2:** Optical images of Popocatépetl volcano acquired before and after the M$_W$ 7.1 Puebla-Morelos earthquake (images ©DigitalGlobe) and landslide map. In the pre-event image (a) acquired on 23 March 2017, the 3800-m line and Los Arenales (LA) deposits are indicated in white. In the post-event image (b) acquired on 11 December 2017, the red arrows indicate the debris flows that occurred in Hutzilac, Hueyatlaco and Xalipilcayatl ravines. In the landslide map (c), the black polygons corresponding to landslide scars and deposits were extracted from a Pléiades 1A image acquired on 13 November 2017. Main volcano-tectonic lineaments are reported in the map (red lines). The white triangle indicates the PPIG seismic station where a PGA of 106.83 cm/s$^2$ was measured (Figure 1c). The black squares indicate the areas zoomed in Figure 3. Background: map of the PGA distribution (data source: USGS Earthquake Hazard Program) and 12.5m DEM ©JAXA/METI ALOS PALSAR 2008 (coordinate system WGS84 UTM Zone 14Q).

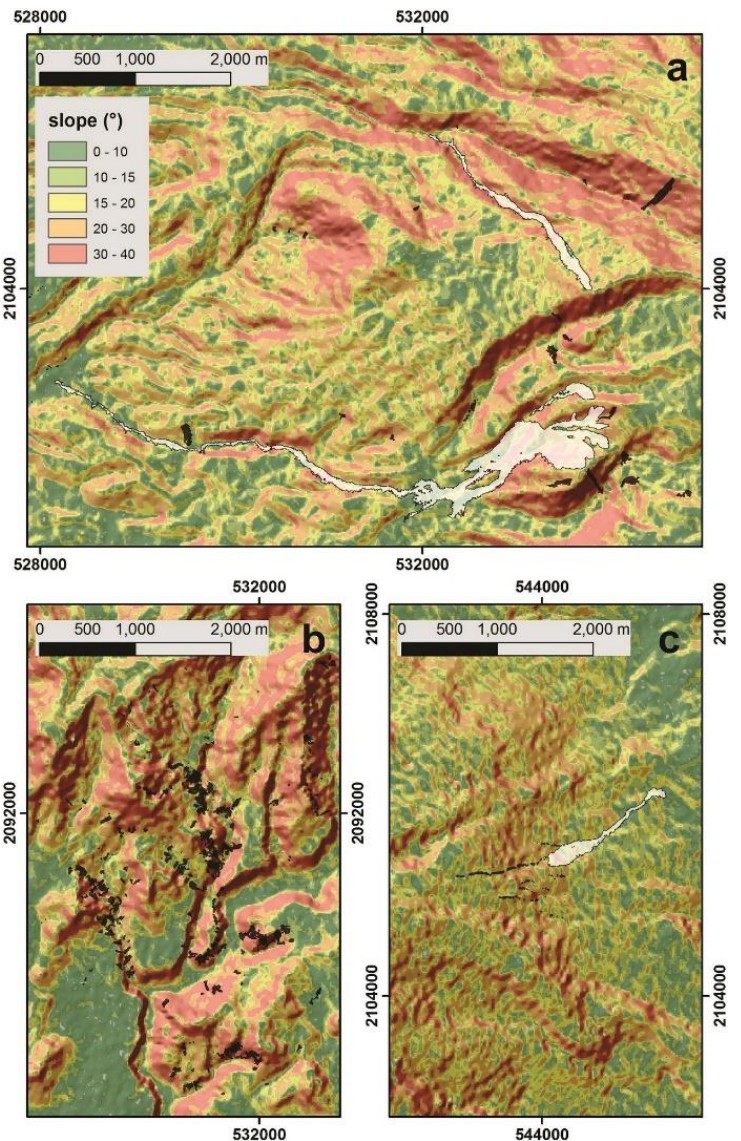

**Figure 3:** Details of the most affected areas of the Popocatépetl volcano by co-seismic landslides. Black polygons correspond to shallow-landslide scars and deposits, whitish areas indicate debris flows. Location of the a, b and c squares are reported in Figure 2. Background: 12.5m DEM ©JAXA/METI ALOS PALSAR 2008 (coordinate system WGS84 UTM Zone 14Q).

We conducted four field campaigns from October 2017 to November 2019 to investigate the morphology and stratigraphy of the source area of main landslides, to map and measure faults and fractures caused by the earthquake, and to define the extension, thickness and textural characteristics of the larger debris flows (Figure 4). The stratigraphy on the main landslide scars was reconstructed to determine texture and physical properties of the tephra layers involved in the mass wasting process. We selected a soil sample for radiocarbon analysis to identify the age of the stratigraphic sequence and to define its

distribution. The $^{14}$C age was obtained through accelerator mass spectrometry dating (BETA Analytic Laboratory) and calibrated with the IntCal20 calibration curve (Reimer et al., 2020). We mapped and sampled main debris-flow deposits and grainsize analysis were performed by dry-sieving for the sand fraction and by means of a laser particle sizer (Analysette 22) for silt and clay fractions.

We analyzed two Sentinel-1 SAR images (Synthetic Aperture Radar, COPERNICUS program) to define the timing between

the earthquake and the observed mass wasting processes (Figure 9). The analyzed images were acquired before and after the earthquake (17 and 23 September 2017) in 1A level Ground Range Detected, ascending orbit, Interferometric Wide sensor mode and dual-polarization. A radiometric calibration was applied to extract the most significant amount of backscattering information from the ground linked to the surficial roughness. As a second step, a change detection technique named Log-Ratio was applied to detect pixel values directly related to radar backscattered correlated to superficial processes; this is an

algorithm used to detect changes using a mean ratio operator between two images of the same area but taken at different times (Mondini, 2017; Singh, 1989). Finally, we analyzed rainfall data gathered at the Altzomoni raingauge station (ALTZ, Figure 1b), located at approximately 10 km north from the volcano summit.

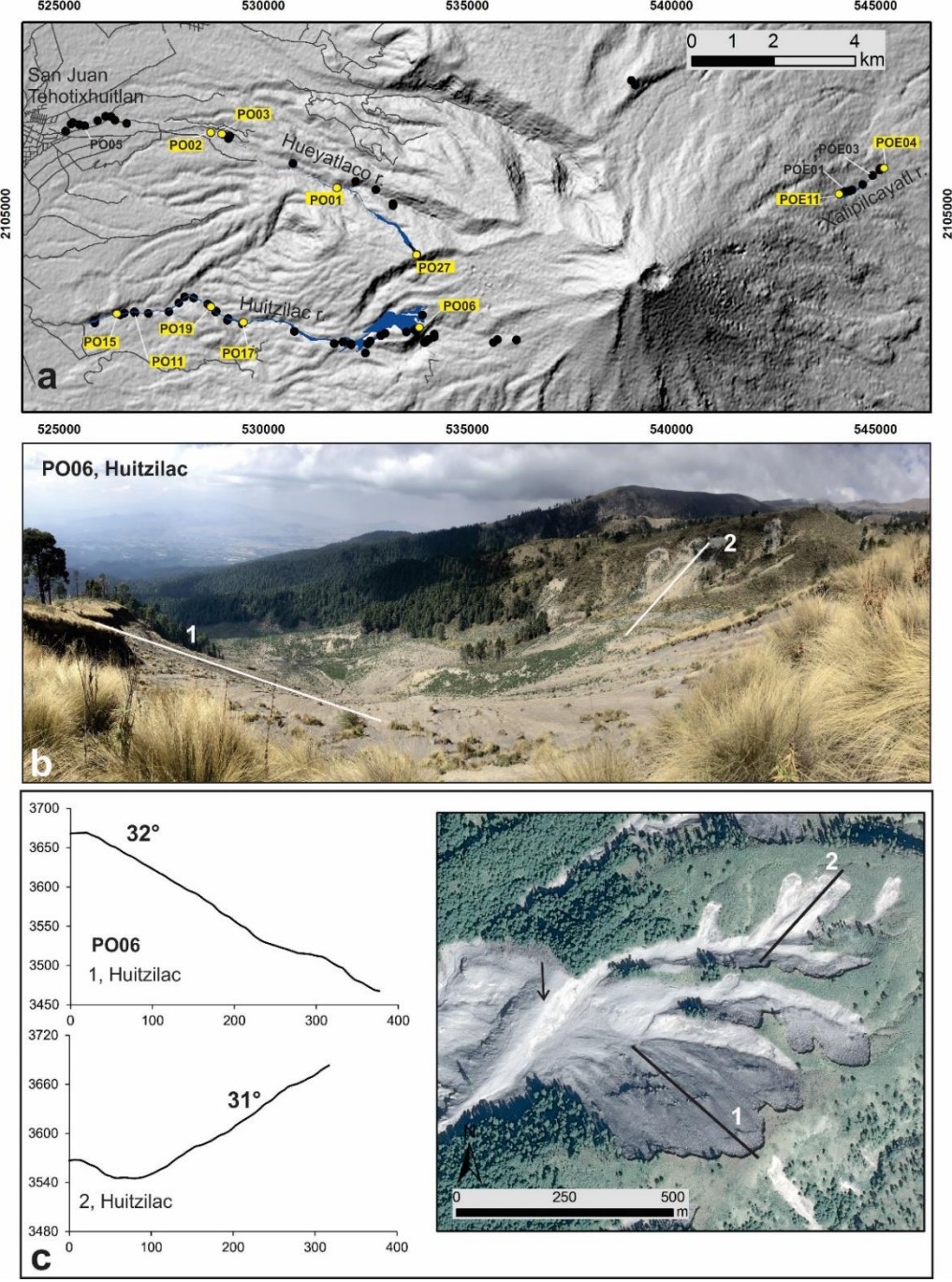

**Figure 4:** (a) Map of three ravines of the Popocatépetl volcano affected by the debris flows; black dots indicate locations of field surveys (background: 12.5m DEM ©JAXA/METI ALOS PALSAR 2008, coordinate system WGS84 UTM Zone 14Q). (b) View in the downstream direction from the top of the sampling point PO06, located on the scarp of the Huitzilac landslide. (c) Topographic profiles of the larger landslides (1 and 2) that occurred at Huitzilac, which are also reported on the Pléiades 1A image acquired on 13 November 2017; the black arrow indicates the topographic barrier overtopped by the debris flow.

## 4 Results

### 4.1 Landslide mapping

The earthquake triggered hundreds of shallow landslides on the volcano slopes covering a total area of 3.8 km² (Figure 2). Soil slips affected the modern soil and part of the unconsolidated volcaniclastic cover. The five largest slope failures occurred in the basins of Hueyatlaco and Huitzilac on the West side of the volcanic edifice, and in the basin of Xalipilcayat on the East (Figure 4). The scarps of these landslides were generated at elevations of about 3400-3800 m a.s.l. on the internal faces of ravines or glacial cirques, where slopes are >20° (Figure 4c). Sharp rectilinear extensional fractures and small normal faults parallel to the valley slopes were observed in the Hueyatlaco basin after the earthquake (Figure 5a). These faults and fractures have maximum length of about 1 km, show displacements of up to 40-50 cm and are located on the valley flanks (Figure 5b), suggesting a correlation with local gravitational instability triggered by seismicity. A cluster of smaller shallow landslides is visible on the southwestern side of the volcanic cone (Figure 2c). These landslides were produced by the collapse of the steep slopes of hummocky hills (Figure 3b) corresponding to the debris avalanche deposit of the last major flank failure that occurred at 23.5 ka PB (Espinasa-Perena and Martín-Del Pozzo, 2006; Siebe et al., 2017).

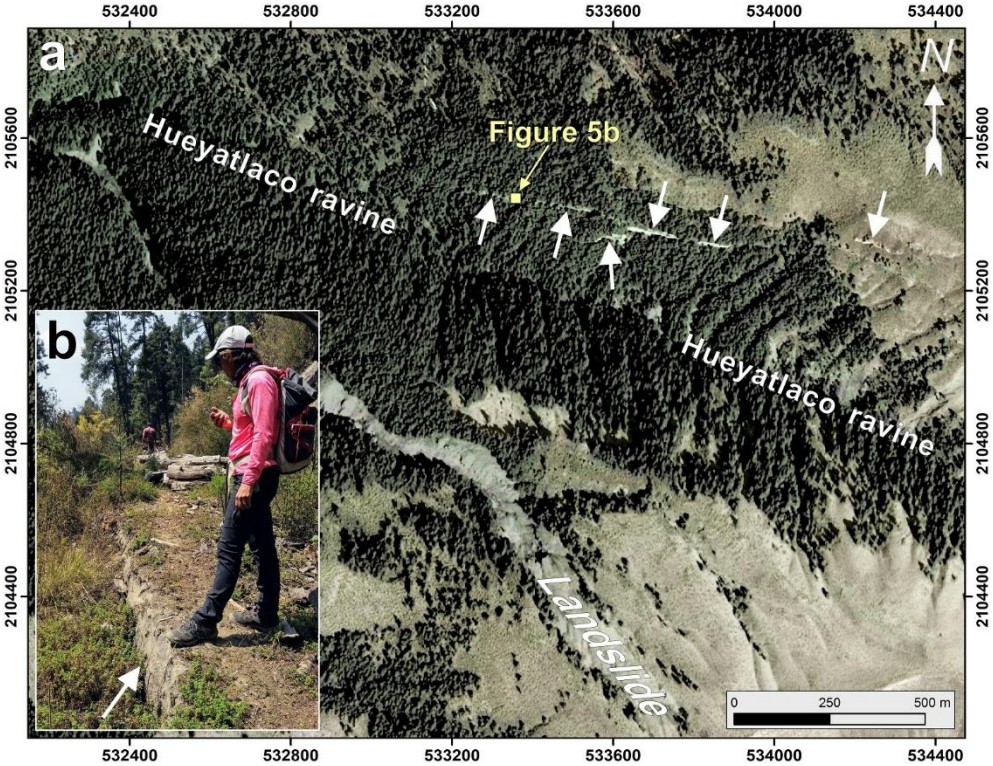

**Figure 5:** (a) Rectilinear extensional fractures and small normal faults opened parallel to Hueyatlaco ravine (white arrows), background image: Pléiades 1A image acquired on 13 November 2017. (b) Detail of the normal displacement of about 50 cm (white arrow). Coordinate system WGS84 UTM Zone 14Q.

The scarps or the larger landslides located on the western slope of the volcano show a similar stratigraphy, with the intercalation of pumice- and ash-fall deposits (Figure 6a-c). Pumice-fall deposits consist of open-framework, clast-supported units composed of gravel-sand sized fragments of pumice embedded in a matrix of fine material (from silt to clay). Two main layers of pumice-fall deposit were observed at the Hueyatlaco and Huitzilac landslide scars (layers B and D, section PO1906; layers C and E, section PO1927, Figure 6e). Another pumice-fall deposit crops out at the base of the pyroclastic succession. The fallout deposits are intercalated with massive o stratified ash layers, with variable thicknesses up to 4 m. They mainly consist of sand (71-93%), silt (16-1%) and less than 1% of clay (see Appendix A). A sample from layer C (section PO06) was dated by using $^{14}$C, giving a calibrated age 537-643 AD ($1500 \pm 30$ BP conventional radiocarbon age) (Figure 6e). Based on this age, the two younger pumice-fall deposits are here correlated with the Upper and Lower Classic Plinian Eruptions (UCPES and LCPES) of the late Holocene, which had main dispersal axis towards E and NE (Figure 1b) (Siebe et al., 1996). The thicker deposits of these eruptions crop out on the eastern flank of the volcano, as observed at section PO11, and correspond to the scar of the Xalipilcayatl landslide (Figure 6d). Here, a main unit of pumice-fall deposit (C in Figure 6d) features a total thickness of 3.5 m, and consists of a massive, clast-supported unit dominated by coarse fragments barren of any silt and clay fractions. This latter unit lies on a 10 cm-thick sandy layer (B in Figure 6d). In all the studied sections, the upper ash unit corresponds to the products accumulated from the frequent vulcanian explosions that characterize the modern eruptive activity of the volcano.

**Table 1:** Main morphometric data of the landslides that occurred in the headwaters of Hueyatlaco, Huitzilac and Xalipilcayatl ravines. The area of the main scars was inferred from the inspection of post-event optical images (see Figure 8). The depth of the scars was measured in the field. The volume of the landslides was calculated assuming a constant depth (with an uncertainty of $\pm0.5$ m) over the area of detachment.

| | Max elevation (m) | Area (m$^2$) | Depth (m) | Slope (°) | Volume × 10$^3$ (m$^3$) |
|---|---|---|---|---|---|
| **Hueyatlaco** | 3 860 | 60 000 | 4 | 29 | 240 ±30 |
| **Huitzilac** | 3 700 | 310 000 | 3 | 31-32 | 930 ±155 |
| **Xalipilcayatl** | 3 500 | 60 000 | 3 | 25 | 180 ±30 |

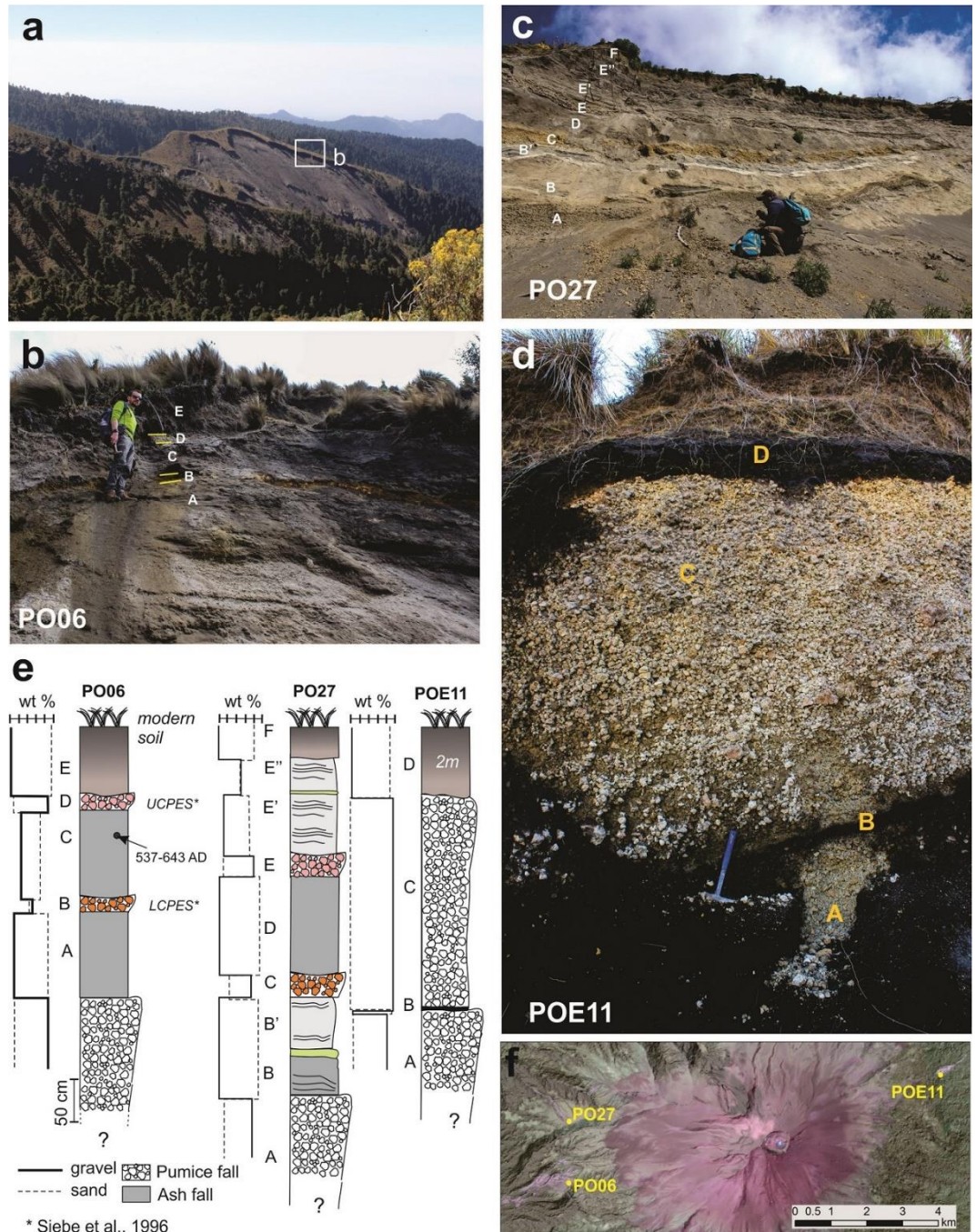

**Figure 6:** View of the three main landslides scarps: (a, b) Huitzilac, (c) Hueyatlaco, (d) Xalipilcayatl. In (e) the stratigraphic sections of the scarps and the grain size distributions of their strata (data in Appendix A) are reported; Upper Classic Plinian Eruptions (UCPES, pink), Lower Classic Plinian Eruptions (LCPES, orange) and the dated layer are indicated, see text for more details. (f) Geographic location of sampling points, background image: Pléiades 1A image acquired on 13 November 2017.

## 4.2 Characterization of debris flows and associated deposits

The five largest landslides described in Section 4.1 (one at Hueyatlaco and Xalipilcayatl, respectively, and three at Huitzilac) mobilized a total volume of about $1.35 \times 10^6$ m³ of ash- and pumice-fall deposits (Table 1). Landslide scarps measured 640 m of length and 4 m of depth at Hueyatlaco, 740 m of length and 3 m of depth at Huitzilac, and 400 m of length and 3 m of depth at Xalipilcayatl (Figure 7). We calculated the volume of the landslides by assuming a constant depth (with an uncertainty of ±0.5 m) over the area of detachment. We measured the depth of the main scars in the field while the area of the main scars was inferred from field surveys and from the inspection of post-event optical images (Figure 7).

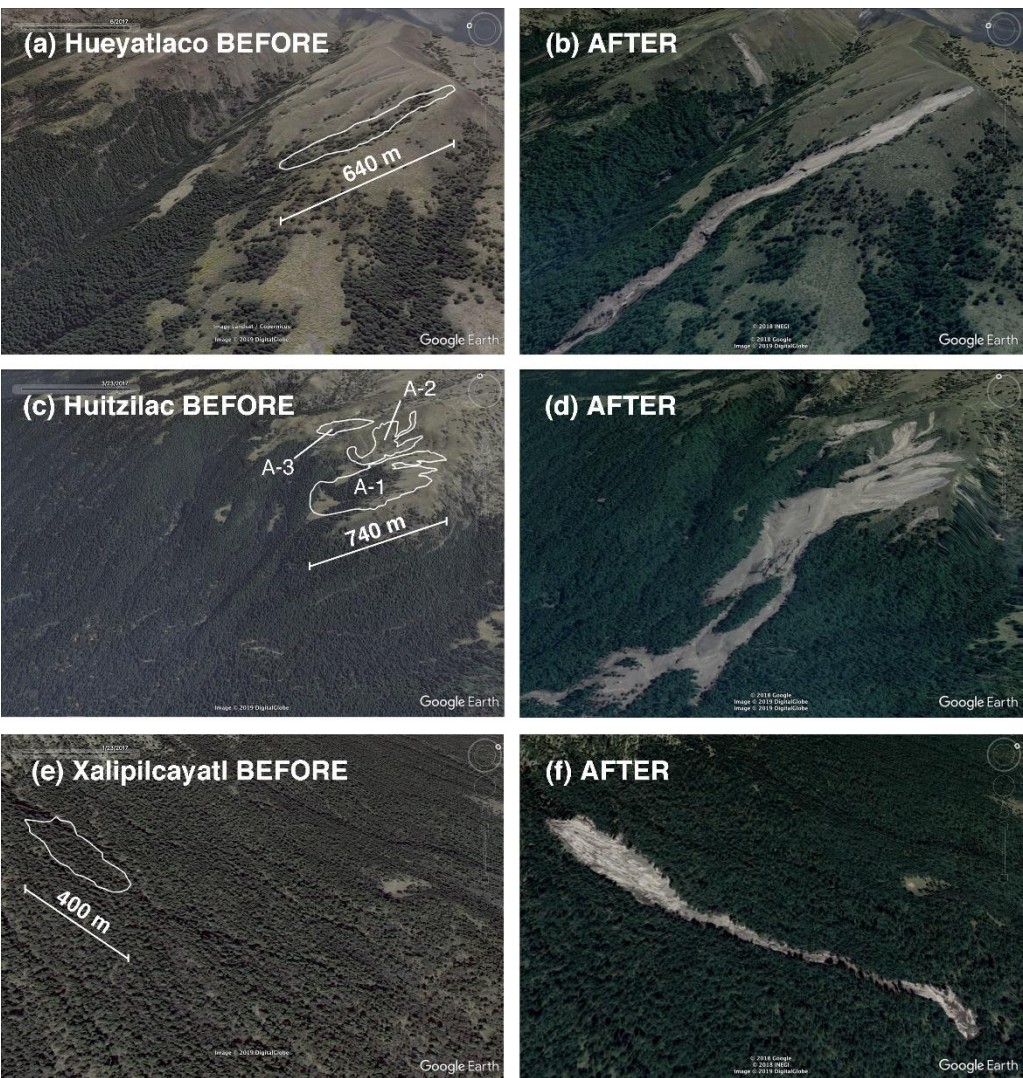

**Figure 7:** Comparison of 3D views on the detachment areas in the headwaters of Hueyatlaco (a,b), Huitzilac (c,d) and Xalipilcayatl (e,f) ravines before and after the larger landslides. Images from Google Earth (©Google 2018, ©INEGI 2018 and ©DigitalGlobe 2019).

The landslides transformed into three long-runout debris flows (Figure 8). At Huitzilac ravine, the main landslide body (landslide A-1, Figure 7c) impacted the opposite side of the valley, partly overtopping it (Figure 4c and 8a). Two other soil slips (landslide A-2 and A-3, Figure 7c) contributed to forming the subsequent debris flow, which extended up to 7.7 km from the source before diluting into a streamflow. The total observed thickness of the deposit measures up to 3 m, but mud traces on standing trees and on lateral terraces measure up to 10 m on proximal reaches (PO17, Figure 8b) and up to 1.5 m in distal reaches with horizontal surfaces at benches (PO11, Figure 8f). In distal reaches, where the channel was shallow, the flow inundated large plains (PO15 and PO19). The deposit is massive, dark-gray in color, and mainly consists of sand (77-86%) with a relevant gravel proportion (15%) due to pumice fragment enrichment in proximal reaches (Figure 8i). Clay content is less than 1%. The lower unit consists of coarse-to-medium ash with evidence of dewatering (Figure 8g). At Hueyatlaco, the debris-flow runout extended up to 6.4 km (Figure 4). The deposit appears as a main unit, dark-gray in color, massive and homogeneous with sand fraction consisting of 70% in proximal reaches (PO01) to 87% in distal reaches (PO05), with up to 15% of silt and less than 1% of clay (Figure 8i, see also Appendix A). Overbank deposits show sharp edges up to 10-cm thick (PO02, Figure 8e). The total observed thickness is up to 50 cm (Figure 8d, erosion was only incipient at the time of the observation) but watermarks up to 5 m were observed in proximal reaches (PO1701, Figure 8c). Finally, the deposit in the Xalipilcayatl ravine extended up to 1.5 km (Figure 7f) and is clearly composed of two main units. The lower unit is massive, dark-grey in color and mostly consists of sand fraction (88%, POE03-lower, Figure 8i ), up to 1.2 m in thickness, while the upper one is massive, pumice-enriched and represents up to 40% of the total unit (POE04, Figure 8h and 8i).

We estimate a total entrainment of about 205 000 $m^3$ along both hillslopes and channel network assuming 0.5 m of erosion over the area located downstream from the main scars (Table 2). Large Wood (LW) elements entrained by the initial landslides and the subsequent debris flows contributed to the final bulk deposits of about $1.632 \times 10^6$ $m^3$. The volume of LW was calculated considering a mean tree height of 25 m (measured in the field, with an uncertainty of ±5 m), a mean trunk diameter of 0.4 m (observed in the field, with an uncertainty of ±0.1 m) and a mean distance of two trees of 10 m (estimated by using the post-event optical images, see Figure 7). The amount of LW recruited in the Huitzilac basin results was 60 000 $m^3$ (±3 000 $m^3$), far more than the sum of wood recruitment estimated for Hueyatlaco (10 000 ±500 $m^3$) and Xalipilcayatl (7 000 ±350 $m^3$) basins. The recruited LW stemmed from the combination of hillslope and channel processes originated from the earthquake-induced landslides. In general, these landslides were the dominant recruitment processes in headwaters. In contrast, LW recruitment from lateral bank erosion became significant in the intermediate reaches of the channels. The slope area that collapsed into the Xalipilcayatl basin contained most of the LWs that was later transported by the flow (86%). In the Huitzilac basin, the LW recruitment mainly occurred on the slopes located right below the collapses (62%), while in the Hueyatlaco basin most came from the channel banks (75%). Most of the transported LWs remained trapped by natural obstacles in the main channel (i.e. standing vegetation) and clogged in the flat reaches of the channel (Figure 8d). In the Xalipilcayatl ravine, most of LW was transported for the whole runout distance into the main landslide deposit.

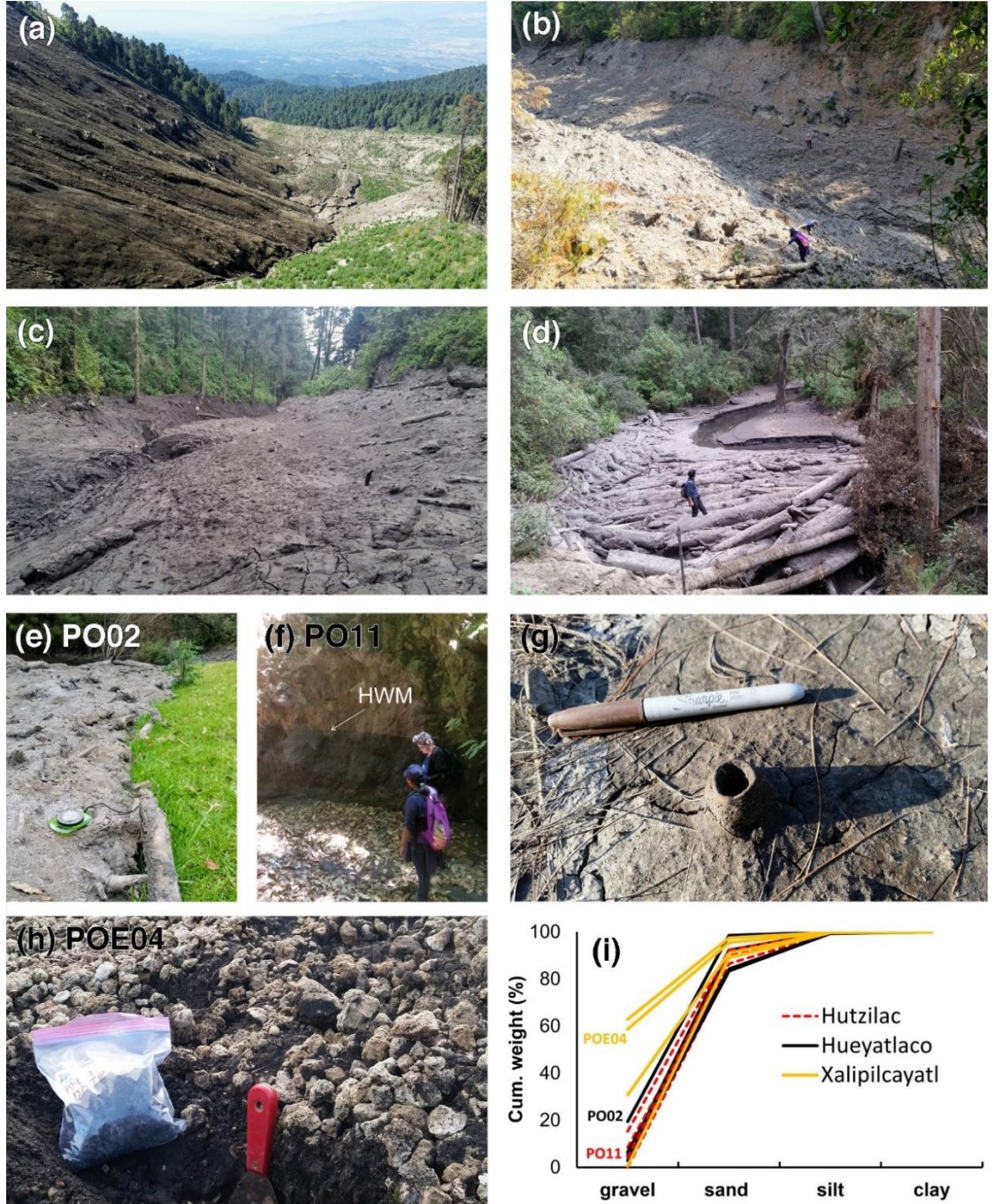

**Figure 8:** Debris-flow deposits in the upper (a-c), intermediate (d-f) and lower reaches (g-h) of Huitzilac, Hueyatlaco and Xalipilcayatl basins: (a) scarp of landslide A-1 at Huitzilac (view from point PO06), (b) main channel of Huitzilac ravine (PO17), (c) main channel of Hueyatlaco ravine (PO1701), (d) large wood deposits at Hueyatlaco (PO03), (e) overbank deposits at Hueyatlaco (PO02), (f) mud trace on lateral terraces at Huitzilac, HWM = Height of Water Mark (PO11), (g) evidence of dewatering at Huitzilac (PO19), (h) detail of the lower deposit at Xalipilcayatl (POE04), (i) grainsize distribution of the samples of the deposits.

**Table 2:** Main morphometric data of the debris flows that were observed in the Hueyatlaco, Huitzilac and Xalipilcayatl basins. The entrained volume was calculated assuming 0.5 m of erosion over the area located downstream from the main scars where the vegetation was destroyed. The volume of large wood (LW) recruitment was calculated considering a mean tree height of 25 m (with an uncertainty of ±5 m), a mean trunk diameter of 0.4 m (with an uncertainty of ±0.1 m) and a mean distance of two trees of 10 m based on field observations and inspection of post-event optical images.

|  | Runout (km) | Drop height (m) | Entrainment × $10^3$ (m³) | LW volume × $10^3$ (m³) |
|---|---|---|---|---|
| **Hueyatlaco** | 6.4 | 1 160 | 50 | 10 ± 0.5 |
| **Huitzilac** | 7.7 | 1 200 | 120 | 60 ± 3 |
| **Xalipilcayatl** | 1.5 | 350 | 35 | 7 ± 0.35 |

### 4.3 Timing of the events

Results of Sentinel-1 SAR image processing clearly indicate that both landslides and debris flows occurred between 17 and 23 September 2017. A binary image was produced where pixels values are linked to spatial change that occurred in this spam of time (Figure 9a). Their distribution corresponds with the deposits of the larger debris flows that occurred in Huitzilac and Hueyatlaco basins, as it is easily observable in a later optical Sentinel-2 image (COPERNICUS program) acquired on 18 October 2017 (Figure 9b).

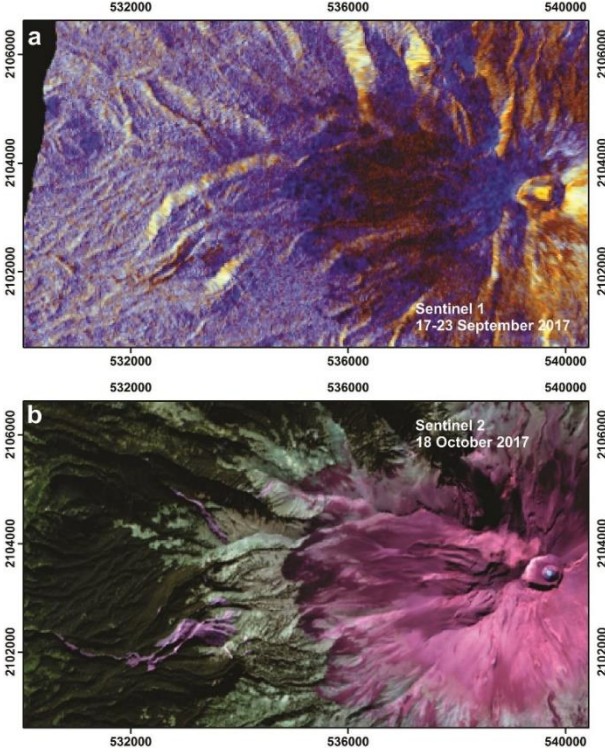

**Figure 9:** (a) RGB (R, post-earthquake; G, pre-earthquake; B, ratio between post- and pre-event) representation of the 17 and 23 September Sentinel-1 (©Copernicus data) change in amplitude analysis; (b) RGB composition of post-event Sentinel-2 image (©Copernicus data). Coordinate system WGS84 UTM Zone 14Q.

A total of 200 mm of accumulated rainfall were recorded during the 30 days preceding the earthquake, with the accumulation of 19.7 mm two days before the earthquake (Figure 10). Thus, we expect that the slope material was wet at the time of the earthquake. Based on the remote sensing analysis and considering that between 19 and 23 September only a few
mm of rainfall accumulated (Figure 10), it is thus highly probable that both slope failures and debris-flow emplacement were co-seismic. Witnesses from the town of Atlautla, which is located at the outlet of Huitzilac ravine (Figure 1b), also confirmed this information. During the following weeks, rainfall remobilized fine material from the landslide deposits reaching the town of San Juan Tehuixtitlán (Figure 4a). On 4 October 2017, the population of San Juan Tehuixtitlán noticed the transformation of the shallow water-flow of Hueyatlaco ravine into a hyperconcentrated flow. It was the first time that
this local community located on the western volcano slope observed such a phenomenon. Rainfall measurement at Altzomoni raingauge station (ALTZ, Figure 1b) shows an accumulation of 35.7 mm of rainfall over 12 hours beginning at 10 (UTC time) of 4 October, with a peak between 20 and 21 (Figure 10). The rainfall event of 4 October only remobilized fine material from the landslide deposits reaching the town of San Juan Tehuixtitlán; the debris flows along the Huitzilac and Xalipilcayatl were never reported since they never extended out to any populated area in 2017. During the 2018 and 2019
rainy seasons, the fine sediment remobilized from the debris-flow deposit in Huitzilac ravine reached the road connecting San Juan Tehuixtitlán to Atlautla (Figure 1b).

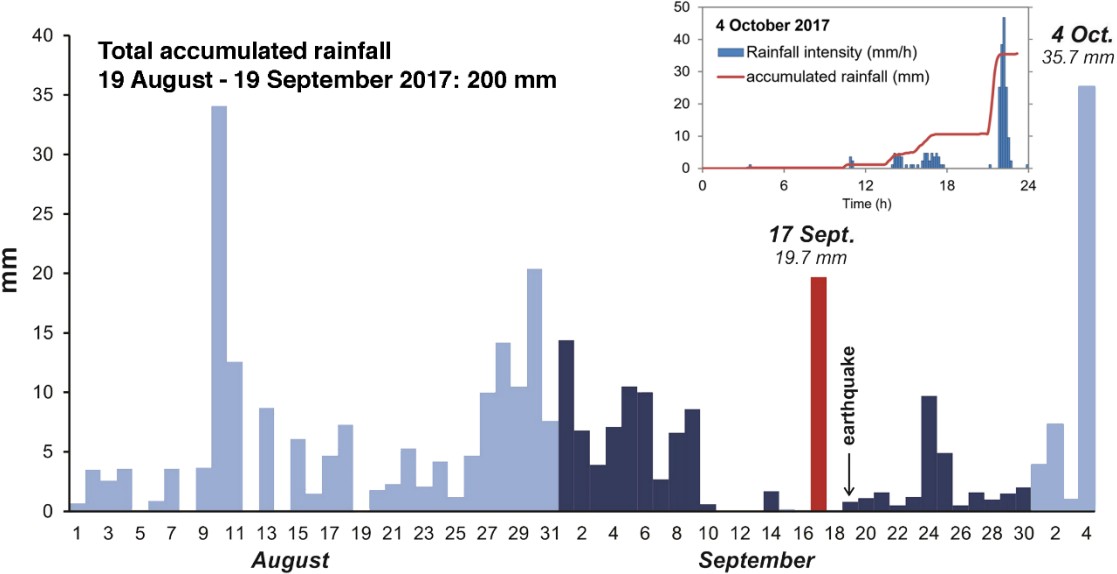

**Figure 10:** Rainfall measurements at rain gauge ALTZ from 1 August to 4 October 2017. A total accumulated rainfall of 200 mm was recorded during the 30 days preceding the earthquake, 19.7 mm of which was on 17 September 2017 (red bar). On 4 October 2017, the
population of San Juan Tehuixtitlán noticed the passage of a sediment-laden flow in Hueyatlaco ravine.

# 5 Discussion

## 5.1 Predisposing factors to slope instabilities

Popocatépetl area is tectonically characterized by a Quaternary roughly NE-SW/ENE–WSW trending maximum horizontal stress regime, responsible for arc-parallel E-W-striking transtensive faults and NE-SW/ENE–WSW arc-oblique normal

faults (Arámbula-Mendoza et al., 2010; García-Palomo et al., 2018; Norini et al., 2006, 2019). This stress regime generated ENE–WSW extensional fracturing and faulting of the volcanic edifice (Figure 11), controlling the orientation and propagation by magmatic overpressure of dikes within the volcanic cone and recent eruptive fissures on its flanks (Arámbula-Mendoza et al., 2010; De Cserna et al., 1988).

The size of the slope failures triggered by the 2017 $M_W$ 7.1 Puebla-Morelos earthquake is highly variable although (i) the

epicenter of the earthquake is far from the volcano, with seismic shaking expected to be of similar intensity all over the symmetric volcanic cone, and (ii) soil and recent pyroclastic cover is quite homogeneous on the edifice flanks. Small shallow landslides occurred all over the volcano flanks, while the few larger landslides described in our work are limited to the eastern and western sides of the volcanic cone (Figure 2). Thus, seismic shaking originated by the earthquake triggered large (volume $> 10^5$ m$^3$) landslides only in specific sectors of the volcano flanks.


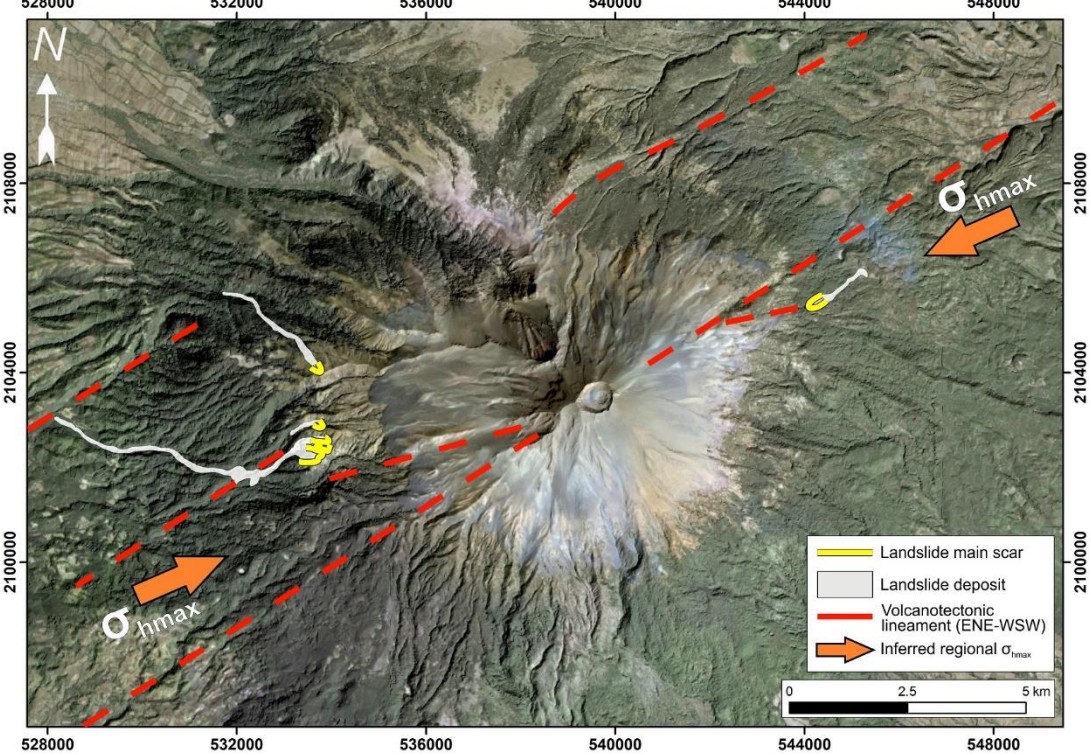

**Figure 11:** Simplified tectonic setting of the Popocatepetl area and location of the main earthquake-induced debris flows that occurred on 19 September 2017. Background image: ©DigitalGlobe/ESRI 2019, coordinate system WGS84 UTM Zone 14Q.

The location of the larger slope failures defines a sharp ENE-WSW unstable sector crossing the volcano summit and parallel

to many deep rectilinear valleys carved in the volcanic cone (Figure 11). In this ENE-WSW elongated sector of the volcano, some faults and extensional fractures have been generated by the 2017 earthquake in the same basins where the larger landslides occurred (Figure 5). This configuration suggests strongly localized site effects and/or a structural control on the location of the slope instability. Indeed, the unstable sector is roughly parallel to the ENE–WSW maximum horizontal stress, where local volcanotectonic structural features are recognized on the volcano (Arámbula-Mendoza et al., 2010; De Cserna et

al., 1988). The remobilization of larger quantities of material in this sector with respect to other areas of the volcano flanks may be correlated to the presence of ENE-WSW–striking faults and fractures that progressively weakened the volcanic edifice. Some of these volcanotectonic structures may also have undergone transient reactivation by seismic shaking, increasing local slope deformation by opening of fractures that promoted the largest slope failures triggered by the earthquake.

**5.2 Initiation of co-seismic landslides**

Slopes collapse when the shear stress across a potential failure plane exceeds the substrate strength. Earthquakes reduce the slope stability and can cause landslides through the perturbation of the normal and shear stresses in the slope. In case of soft, saturated soils, the coalescence of cracks during earthquakes may results in liquefaction due to the increase of substrate permeability. At Popocatépetl volcano, a combination of these two mechanisms produced the soil slips observed in the

headwaters of Hueyatlaco, Huitzilac, and Xalipilcayatl basins. Shapiro et al. (2000) already noticed that a large earthquake occurring in the vicinity of the volcano may result in flank instability because of the seismic waves traversing the poorly consolidated material composing the volcanic edifice. The ground motion during the 2017 earthquake was anomalously large in the frequency range 0.4–1 Hz, as intraslab earthquakes involve higher stress drop than their interplate counterparts (Singh et al., 2018). Consequently, the ground motion is relatively enriched at high frequencies as compared with that during

interplate earthquakes, which is dominated by lower frequency waves (f< 0:5 Hz), and this effect can contribute to explaining the high value of PGA measured on the volcano slope.

Unexpected large peak accelerations have been recorded along crests of mountain ridges during several earthquakes (Davis and West, 1973; Meunier et al., 2008). Topographic amplification of ground vibrations is primarily due to the reflection/diffraction of seismic waves, which are progressively focused upwards (Bouchon et al., 1996; Davis and West,

1973). The constructive interference of reflections and the associated diffractions of seismic waves increase towards the ridge crest also due to local geologic factors, giving rise to enhanced ground accelerations on topographic highs (Del Gaudio and Wasowski, 2007; Meunier et al., 2008; Von Specht et al., 2019). Geli et al. (1988) show that topographic complexity (presence of neighboring ridges) may be responsible for large crest/base amplifications resulting in complex amplification-deamplification patterns and significant differential motions along the slopes. The amplification at the crest of a mountain

can be as large or larger than the amplification normally caused by the presence of near-surface unconsolidated layers (Davis and West, 1973). It is well-known that shallower earthquakes may cause large landslides (e.g., Marc et al., 2019), but the

Puebla-Morelos earthquake was moderately deep (i.e., 57 km). The PGA produced by the 2017 earthquake at station PPIG (106.83 cm/s$^2$) was about two times higher than the PGA observed at CU (57.1 cm/s$^2$). Indeed, the distance epicenter-PPIG (68 km) is about half than the distance epicenter–CU (111 km) and this partially explains the difference in PGA observed at the two stations. However, during the earthquake the headwaters of Hueyatlaco, Huitzilac, and Xalipilcayatl ravines could have experienced even higher values of PGA due to the effect of topographic amplification of seismic waves. The PGA map produced by the USGS Seismic Hazard Program shows values between 0.28$g$ in the southern sector of the cone and up to 0.18$g$ closer to the vent (Figure 2c). The spatial interpolation of PGA clearly shows the interaction between the energy distribution and the topography, which played an important role in the location of landslides. The cluster of smaller landslides located on the southwestern side of the volcanic cone, closer to the epicenter, is likely due to the combination of large ground motion and high slopes that consist of debris avalanche hummocks (Figure 3b).

The complex topography of Popocatépetl volcano, characterized by neighboring ridges and valleys, probably produced local amplification values that makes it difficult to explain why larger soil slips did not occur in other similar locations in terms of elevation, slope and stratigraphy. However, the deposits located along the ENE-WSW unstable sector of the volcano (see section 5.1), at an elevation ranging from 3400 to 3800 m and characterized by a slope > 20°, appear as the most likely to suffer collapse in case of an earthquake. This sector of Popocatépetl volcano consists of a mantle of loose volcaniclastic material with the intercalation of silty-sandy ash layers and gravel-sand pumice fall deposits (up to 5-m thick, see Figure 6), covered by a modern soil with thick alpine grassland. At higher altitude, the steeper slopes are unvegetated, and consist of unconsolidated pyroclastic granular material where superficial granular flows can be easily observed. The largest landslides occurred in the limit of the vegetation line, where pine tree became scattered but grassland is still abundant (Figure 7a, c). The intercalation of layers with different grainsize and the soil coverage are probably promoting water accumulation. Indeed, one mechanism that possibly can explain the collapse of this material is liquefaction through the disruption of internal, suspended aquifers. A similar observation was recently made at Nevado del Huila Volcano, Colombia, during 2007 when lahars originated after large fractures formed across the summit area of the volcano in consequence of a strong hydromagmatic explosion that drained small, perched aquifers (Johnson et al., 2018). On the unvegetated portion of the cone, mass remobilization processes such as raveling and superficial granular flows likely occurred but without leaving any scarp, because of the lack of a compacted soil.

### 5.3 Transformation into long runout debris flows and implications for hazard assessment

Once generated, the earthquake-induced soil slips transformed into debris flows. The two major debris flows that occurred in Hueyatlaco and Huitzilac basins covered a runout distance of 6.4 and 7.7 km, respectively. In Figure 12, we show the conceptual model of this transformation at Popocatépetl volcano: the propagation of an earthquake-induced crack in the saturated slope (1) produces a shallow landslide composed of a mix of ash and pumice (2). The collapsed material disaggregates and impacts on the opposite side of the valley and rapidly the landslide evolves into debris flows, due to the high-water content of the collapsed unconsolidated material (3). The subsequent debris flow is highly viscous due to the high

sand and silt fraction of the mixture (Figure 8i) and contains abundant LW entrained along the channel network, especially along the Huitzilac and Xalipilcayatl channels which had entire mature trees incorporated, thus leaving abundant log-strewn debris. Even if no direct observations are available to assess whether the collapsed slopes were partially or completely saturated, it is clear that debris flows contained a large amount of water as observed from dewatering features of the deposits and high-water marks along the channels (Figure 8g). Beginning 21 August 2017, 138 mm of rainfall accumulated

continuously for two weeks, with 19.7 mm just two days before the earthquake (Figure 10). This large amount of rainfall was then stored in the open-framework pumice fall deposits intercalated by m-thick sandy layers, and in the root fabric of the trees in the dense forest cover.

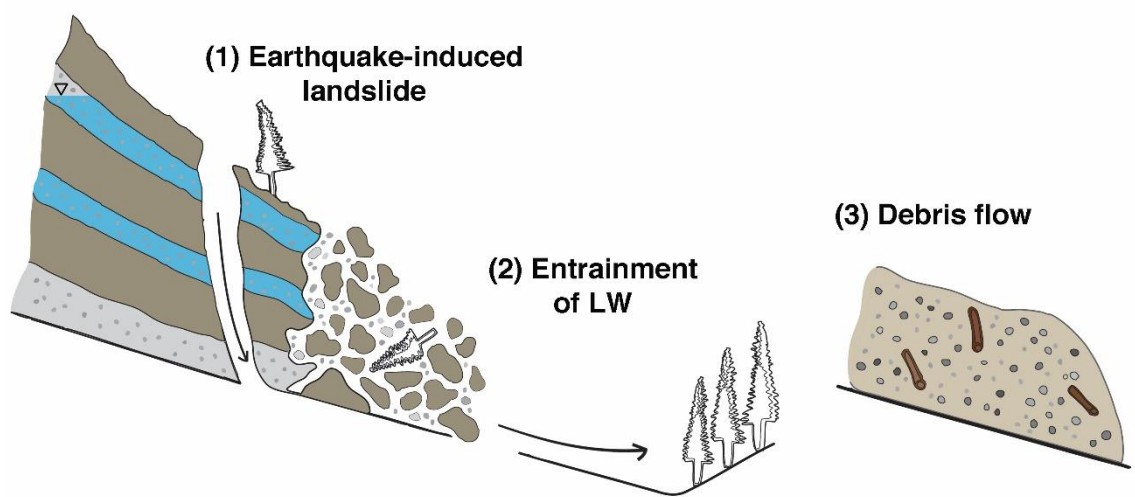

**Figure 12:** Conceptual model of transformation of earthquake-induced soil slips into debris flows at Popocatépetl volcano: (1) the
earthquake produces the collapse of the saturated slope composed of a mix of ash and pumice; (2) the landslide impacts on the opposite side of the valley entraining large amount of Large Wood (LW) and (3) evolves into a debris flow due to the high-water content of the material. The simplified stratigraphy in (1) reflects the one observed at the scarp of Huitzilac landslide (see Figure 6b).

Volcanoes store or drain water in and through aquifers that can grow and empty, as impermeable barriers develop or as they
are breached by deformation, respectively (Delcamp et al., 2016). Even if not completely saturated, ground vibrations induced positive pore pressure and triggered liquefaction and slope failure (Kameda et al., 2019; Wang et al., 2019). It is important to note that on August 10 a rainfall of 35 mm, similar to the October 4 event that triggered the sediment-laden flow observed at San Juan Tehuixtitlán village, did not induce any channel response, indicating the stability of the slopes of this sector of the volcano prior to the earthquake. In fact, except for the 2010 lahar that occurred in the Nexplayantla ravine
after 100 mm of accumulated rainfall (Zaragoza-Campillo et al., 2020), lahars are related to major eruptions, which explains why the hazard map of the Popocatépetl volcano includes only rainfall-triggered lahar during or after eruptions (Martin Del Pozzo et al., 2017). Detailed field investigations of the role of aquifers on volcanic landslides are very scarce to the date

(Delcamp et al., 2016). Knowledge of the distribution of perched aquifers and water content of volcanic deposits can provide precious insights into a complex mass wasting chain like the one that experienced by Popocatépetl volcano in 2017.

Finally, it is worth mentioning that during our last field campaign of November 2019 we observed that the source areas of the larger debris flow that occurred at Hutzilac ravine were becoming stable as a result of the combined effect of the removal of fine material and of the growth of new vegetation. On the contrary, the large amount of material deposited in the channel remains available for remobilization for many years, resulting in a remarkable increase of sediment yield as observed in other locations (e.g., Fan et al., 2021). Indeed, during 2018 and 2019 rainy seasons, the fine sediment remobilized from the

debris-flow deposits in Huitzilac ravine reached the road connecting San Juan Tehuixtitlán to Atlautla.

## 6 Conclusions

The catastrophic event of 19 September 2017 at Popocatépetl volcano is an exemplary case of interrelated multiple hazards in volcanic environment: earthquake, landslides, and sediment-laden flows. During the $M_W$ 7.1 Puebla-Morelos intraslab earthquake, hundreds of shallow landslides were triggered on the volcano flanks. The combination of strong ground motion

due to local amplification with the presence of water-saturated, tephra-rich superficial deposits resulted in large slope failures and subsequent liquefaction of the collapsed material. A total volume of about $10^6$ cubic meters of volcaniclastic deposits transformed into two large debris flows on the western slope of the volcano and one on its eastern side. While the source areas rapidly stabilize in the months and years following, the fine material deposited in the channels remains exposed to possible remobilization for many years. These observations imply the need to revise the hazards assessment for

Popocatépetl volcano, where multi-hazard risk scenarios should be taken into account, as well as in other volcanic settings. The mass-wasting cascade described here may occur in other locations, especially in continental volcanic arcs and mountain chains located in tectonically active regions.

*Acknowledgements*. This research is supported by the CONACYT-PN 360, PAPIIT-DGAPA 106419 projects and the EARFLOW project (2018-2021) funded by the MAECI - Ministero degli Affari Esteri e della Cooperazione Internazionale
and the AMEXCID - Agencia Mexicana de Cooperación Internacional para el Desarrollo. Seismic data gathered at station PPIG were provided by the Servicio Sismológico Nacional (SSN - UNAM, México). Rainfall data recorded at Altzomoni station (Red Universitaria de Observatorios Atmosféricos - UNAM) were kindly provided by Adolfo Magalli. The article processing charge is supported by the Open Access Publishing Fund of the Free University of Bozen-Bolzano. We thank Berlaine Ortega-Flores, Lizeth Cortez and Lizeth Caballero-García for their support in the field and in the laboratory. We are
thankful for constructive feedback from Thomas Pierson on an earlier version of the manuscript. We thank Matteo Roverato, three anonymous reviewers and the associate editor Xuanmei Fan for their detailed comments and revisions.

*Data availability*. Samples collected on landslide scars and deposits are stored at CGEO-UNAM. Original data presented in Table 1 and 2 (polygons of the main landslides and LWs) and grain size distribution of the samples are available as supplementary material. The map of the PGA distribution of the $M_W$ 7.1 Puebla-Morelos earthquake (USGS Earthquake
Hazard Program) is available at https://earthquake.usgs.gov/earthquakes/eventpage/us2000ar20/shakemap/pga. Sentinel-1 and Sentinel-2 images (©Copernicus data) acquired on 17 and 23 September 2017 are available at https://scihub.copernicus.eu/. Rainfall data recorded at Altzomoni (ALTZ, RUOA-UNAM) are available at

. The 50-cm resolution Pléiades (AIRBUS) optical image acquired on 13 November 2017 was bought by the authors.

*Author contributions*. VC and LC conceived the idea, planned the field activities and collected most of the data. VC, LC, GN, DF and EP participated to the field work. VC, LC and GN wrote the manuscript. VM analyzed seismic data of the PPIG station (SSN-UNAM). ND processed Sentinel-1 and Sentinel-2 images. EP, LC and VC analyzed the post-event Pléiades image and drew the landslide map. All the authors discussed the results and commented on the manuscript.

*Competing interests*. The authors declare that they have no conflict of interest.

**Appendix A**

Grain size distributions of samples collected in the landslide scarps and deposits, cutoff particle sizes: gravel 64 mm – 2 mm, sand 2 mm - 64 micron, silt 64 - 2 micron, clay <2 micron. Refer to Figure 4 for sample locations.

| **Landslide scars** | | | | |
|---|---|---|---|---|
| **PO06** | **Gravel (wt %)** | **Sand (wt %)** | **Silt (wt %)** | **Clay (wt %)** |
| *E* | 3.49 | 93.14 | 3.29 | 0.09 |
| *D* | 87.01 | 11.94 | 1.00 | 0.05 |
| *C* | 26.47 | 71.90 | 1.61 | 0.03 |
| *B* | 54.36 | 45.12 | 0.51 | 0.01 |
| *A* | 13.92 | 81.00 | 4.95 | 0.13 |
| **PO27** | | | | |
| *F* | 0.36 | 82.18 | 16.78 | 0.68 |
| *E''* | 45.75 | 53.30 | 0.91 | 0.04 |
| *E'* | 10.25 | 76.50 | 12.73 | 0.52 |
| *E* | 80.98 | 16.39 | 2.45 | 0.17 |
| *D* | 1.49 | 87.28 | 11.12 | 0.10 |
| *C* | 71.66 | 27.88 | 0.45 | 0.01 |
| *B'* | 0.10 | 83.89 | 15.31 | 0.70 |
| *B* | 0.50 | 83.77 | 15.39 | 0.33 |
| A | 80.06 | 12.40 | 6.98 | 0.57 |
| **POE11** | | | | |
| D | 0.53 | 88.55 | 10.66 | 0.26 |
| C | 94.81 | 5.19 | 0.00 | 0.00 |
| B | 4.12 | 91.84 | 3.87 | 0.16 |
| A | 89.16 | 10.37 | 0.43 | 0.04 |
| **Debris-flow deposits** | | | | |
| *Hueyatlaco ravine* | | | | |
| PO01 | 17.98 | 69.20 | 12.22 | 0.61 |
| PO02A | 19.49 | 79.05 | 1.37 | 0.09 |
| PO02B | 5.65 | 86.42 | 7.78 | 0.15 |

| | | | | |
|---|---|---|---|---|
| PO02C | 4.76 | 87.52 | 7.56 | 0.16 |
| PO05A | 6.46 | 77.81 | 15.19 | 0.54 |
| PO05B | 2.91 | 80.49 | 15.95 | 0.66 |
| *Huitzilac ravine* | | | | |
| PO11 | 4.25 | 86.21 | 9.19 | 0.34 |
| PO15 | 0.09 | 83.54 | 15.82 | 0.52 |
| PO17 | 15.35 | 77.02 | 7.41 | 0.20 |
| PO19 | 8.49 | 77.96 | 13.15 | 0.38 |
| *Xalipilcayatl ravine* | | | | |
| POE04 | 58.75 | 36.69 | 4.25 | 0.30 |
| POE01 | 30.77 | 60.26 | 8.59 | 0.36 |
| POE03-upper | 62.93 | 34.45 | 2.41 | 0.20 |
| POE03-lower | 0.69 | 88.32 | 10.63 | 0.34 |

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
