# Peer review of "Earthquake-induced debris flows at Popocatépetl Volcano, Mexico"

_Earth Surface Dynamics, 2020_

## Short Comment (SC1) · 8 Jun 2020

The paper by Veio Coviello et al. titled "Earthquake-induced debris flows at Popocatépetl Volcano, Mexico" is a tiny substantial and interesting contribution to scientific progress within the scope of Earth Surface Dynamics. The paper addresses relevant scientific questions and presents novel data applying multiple-approach methods to resolve a scientific issue. In the manuscript, it is stressed that those kinds of events have never been described before at Popocatepetl volcano. The paper is generally well written although It could be slightly improved. I'm not an English mother-language speaker, then I suggest an extra check for the English grammar. Indeed, I've found some minor grammatical errors in the text that should be considered. Methods and assumptions are valid and clearly outlined. The manuscript

reveals a commitment to the research presented and a strong field-based work that adds value to the topic. Lab data and computer elaborations of satellite images are provided and well presented. Results discussed are presented in an appropriate and balanced way. References are proper for the presented manuscript as well as the title and the number of figures. Figures and figure captions should be more precise as I mentioned in the attached PDF version. In general, I suggest minor to moderate revisions. With kind regards, Matteo Roverato

Please also note the supplement to this comment:
https://esurf.copernicus.org/preprints/esurf-2020-36/esurf-2020-36-SC1-supplement.pdf

**Supplement:**

[revised manuscript text omitted]

---

## Referee Comment (RC1) · Anonymous Referee #1 · 27 Aug 2020

In this manuscript, the authors analyzed the earthquake-induced landslides/debris flow at Popocatépetl Volcano, Mexico in detail. In the surrounding area of volcano, the general risk is mainly originated from the lahars during the volcano eruption. The mass movement of deposits triggered by earthquake is interesting and could provide some insights to the risk evaluation in the volcano region. Nevertheless, a major improvement is still necessary before its consideration of acceptance on evaluation of its current qualities. First of all, the title of this manuscript is somewhat confusing. The mass flow directly triggered by earthquake is slope failure or landslides. Because of the mass saturated with water or added by water, it could transfer into the debris flow and move further. In current status, the authors didn't clearly describe how the landslide transfer to debris flow. The reason why it can move a long distance came from the initial

acceleration by earthquake or the abundant water content? Thus, the authors should add a new section to clearly introduce the transition process. For this manuscript, the detailed investigation and analysis is an important factor that the readers would be interested. Thus, it is encouraged to provided the original data which the other researchers could conveniently use, especially the initial mass and deposit mass range and depth etc. (KML or ascii files would be excellent) Please rewrite the abstract and clearly describe the findings and interesting point. Now too much detailed description. Line 20: The debris flows were highly viscous and contained abundant large woods (about 10ˆ5cubic meter). This sentence is not clear. The volume refers to the debris flow or woods? The quality of figure 1 a, b is not good. Please try to improve the their performance. The slope direction with woods of the figure 12(2) would be better to exchange.

---

## Referee Comment (RC2) · Anonymous Referee #2 · 6 Sep 2020

General comments: Coviello et al. 2020, entitled "Earthquake-induced debris flows at Popocatépetl Volcano, Mexico" details the co-seismic landslides triggered by the M71 2017 Mexico earthquake and the subsequent debris flows induced by the succeeding rainfall. The manuscript presents a well-planned out geomorphological study comprising of remote sensing analysis, field investigations, grain size analysis of samples collected from the mass movements, and interpretations explaining the context, failure mechanisms, and characteristics of the co-seismic landslides and debris flow. However, I do have some significant suggestions which would be helpful for the authors to improve the presentation of their data and readability of the manuscript. I would suggest minor to moderate revisions following the comments below (also the same is attached as a PDF to ease the revision process). Specific comments: 1.

[Figure]

Focus: The manuscript is entitled "Earthquake-induced debris flows" but analyses both co-seismic landslides and subsequent debris flows induced by the rainfall. I would suggest the authors rethink the title as Mass movements at Popocatépetl Volcano, Mexico induced by the 2017 M7.1 Puebla-Morelos earthquake". 2. Abstract: The focus and the contents of the manuscript are not clear from the abstract. The authors should emphasize that the cause of debris flows post-earthquake is also the rainfall that occurred between 17th to 19th September 2017. Further, antecedent rainfall would also be a key factor to destabilise the volcanic ash and pumice slopes with partial or complete saturation. The predisposing and initiation conditions detailed in the discussion part would better be reflected in the abstract as well. 3. Introduction: The authors introduced the earthquake-induced mass movements in general and provided a brief review of co-seismic mass movements in volcanic regions but only within Mexico or near the study area. As earthquake-induced volcanic mass movements occur in many other places i.e. during the 2018 Hokkaido Eastern Iburi Earthquake and the 2004 Niigata Chuetsu earthquake etc., I suggest the authors provide a paragraph briefly reviewing the earthquake-induced volcanic mass movements occurred elsewhere in the world similar to the 2017 M7.1 Puebla-Morelos earthquake. Please refer to (Sassa 2005; Yamagishi and Yamazaki 2018; Wang et al. 2019) 4. Factors (predisposing): The authors discussed the predisposing factors of the co-seismic landslides in a better way. I would also suggest the authors prepare and complete inventory of the landslides and perform a factor analysis in a GIS environment (would be too much for this manuscript but just a suggestion). By doing this the predisposing factors can be understood statistically including tectonism, earthquake acceleration, geology, stratigraphy, faults, distance to ridges and soil cover (if possible), etc. However, this is just a suggestion and not necessary to include them within this manuscript. 5. Failure mechanism: I commend the authors for the deep thoughts regarding the failure mechanism of pumice fall and volcanic ash deposits during an earthquake. By seeing the antecedent rainfall occurred before the earthquake, it is obvious that partial or complete saturation of pumice deposits are possible. Liquefaction of pumice deposits

is a well-known phenomenon. To elaborate more on the same, I recommend the authors discuss the failure mechanism compared to the case on the 2018 Hokkaido Eastern Iburi earthquake. Wang et al. (2019) and Kameda et al. (2019) provide detailed explanations of the fluidized landslides over pumice deposits. Further, as also suggested by Referee #1, please explain how the co-seismic landslide deposits were transformed into debris flows. Minor specific comments: 1. Line 48: Please mention here how the debris flows were transformed from landslides? With the help of antecedent and subsequent rainfall? 2. Line 65: Figure 1. Please add the coordinates to Figure 1(a) and (b). The units and fonts in Figure 1(c) should be magnified. Overall, please improve this figure. 3. Line 135: Figure 3. Please see is it possible to differentiate the co-seismic landslides and debris flow in this figure? 4. Line 218: Table 2: Would it be better to show the grain-size in a figure? 5. Line 361: Figure 12: Please see, if you can include the effect of rainfall and partial saturation in this conceptual diagram? Technical corrections: 1. Grammatical errors and some spelling mistakes are spotted here and there. Please check them thoroughly. 2. Figure 12: Please correct the spelling for debris flows in (3). Suggested references Kameda, J., Kamiya, H., Masumoto, H., Morisaki, T., Hiratsuka, T. & Inaoi, C. 2019. Fluidized landslides triggered by the liquefaction of subsurface volcanic deposits during the 2018 Iburi–Tobu earthquake, Hokkaido. Scientific reports, 9, 13119, doi: 10.1038/s41598-019-48820-y. Sassa, K. 2005. Landslide disasters triggered by the 2004 Mid-Niigata Prefecture earthquake in Japan. Landslides, 2, 135-142. Wang, F., Fan, X., Yunus, A.P., Siva Subramanian, S., Alonso-Rodriguez, A., Dai, L., Xu, Q. & Huang, R. 2019. Coseismic landslides triggered by the 2018 Hokkaido, Japan (M w 6.6), earthquake: spatial distribution, controlling factors, and possible failure mechanism. Landslides, 1-16. Yamagishi, H. & Yamazaki, F. 2018. Landslides by the 2018 Hokkaido Iburi-Tobu Earthquake on September 6. Landslides, 15, 2521-2524, doi: 10.1007/s10346-018-1092-z.

Please also note the supplement to this comment:

https://esurf.copernicus.org/preprints/esurf-2020-36/esurf-2020-36-RC2-supplement.pdf

---

## Author Comment (AC1) · 4 Oct 2020

We thank Dr Roverato for his support. We revised figures and figure captions as suggested and we performed a general revision of the English. We considered all the minor comments provided to the text, we thank the referee for his detailed reading of the manuscript. Please find the manuscript enclosed with track changes.

Best regards, Velio Coviello

Please also note the supplement to this comment:
https://esurf.copernicus.org/preprints/esurf-2020-36/esurf-2020-36-AC1-supplement.pdf

[Figure]

**ESurfD**

---

## Author Comment (AC2) · 4 Oct 2020

We thank the referee for his constructive comments and suggestions.

We agree, the mass movements directly triggered by earthquake are slope failures and the water content has an important role in the transformation of those landslides in debris flows. However, the Puebla-Morelos earthquake is the main factor controlling this mass-wasting chain. This is the main reason why we would prefer keeping the original title, short and concise. In addition, in this paper we focus on the larger landslides and on their transformation into debris flows.

Concerning the transformation of the main landslides into debris flows, we discussed these topics based on the available data in the source area, which are very scarce.

[Figure]

We added some additional discussion about the water content of the deposits to the section 5.3 Transformation into long runout debris flows and implications for hazard assessment: "Volcanoes store or drain water in and through aquifers that can develop and empty, as impermeable barriers develop or as they are breached by deformation, respectively (Delcamp et al., 2016). Even if not completely saturated, ground vibrations induced positive pore pressure and triggered liquefaction and slopes failure (Kameda et al., 2019; Wang et al., 2019). [. . .] Detailed field investigations of the role of aquifer on volcanic landslides are very scarce to the date (Delcamp et al., 2016). Knowledge of the distribution of perched aquifers and water content of volcanic deposits can provide precious insights on a complex mass wasting chain like the one that experienced Popocatépetl volcano in 2017." Based on our findings, new studies on the hydrology of the volcano are surely needed.

Figure 1 was improved: we added to the map the northamerican plate and the cocos/rivera plates to Figure 1a, we indicated the location of Mexico City and we put the geographical coordinates in Figure 1b. Also, we modified Figure 12 adding some graphical elements indicating the presence of water in the pumice deposits. Finally, we'll provide the original data presented in Table 1 (elevation, area, and depth of the main landslides) as a supplementary material.

Please find the manuscript enclosed with track changes.

Best regards, Velio Coviello

Please also note the supplement to this comment:
https://esurf.copernicus.org/preprints/esurf-2020-36/esurf-2020-36-AC2-supplement.pdf

―――――――――――――――――

**Supplement:**

[revised manuscript text omitted]

---

## Author Comment (AC3) · 4 Oct 2020

We thank the referee for his support.

Similarly to what we replied to referee#1, we agree on the fact that "mass movements" would be more appropriate to describe the entire mass wasting chain that occurred at Popocatépetl in 2017. However, in this paper we focus on the larger landslides and on their transformation into debris flows that represent an exceptional phenomenon in terms of size and runout and, thus, we would prefer to keep the original title.

We now mention in the abstract that remobilized deposits were saturated by heavy rainfall that occurred between 17th to 19th September 2017.

We also thank the reviewer for the valuable insightful references he suggested. We now

refer to the papers describing the effect of the 2018 Hokkaido Eastern Iburi Earthquake both in the introduction and in the discussion. In the revised manuscript, we also mention the similarity in terms of liquefaction observed during this latter earthquake (see section section 5.3 Transformation into long runout debris flows and implications for hazard assessment).We do not provide a complete, global review of the earthquake-induced volcanic mass movements because we feel that it would be a bit far beyond the scope of our work which is focused on the long-runout debris flows observed at Popocatépetl in 2017. Concerning the predisposing factors of the co-seismic landslides, a GIS analysis of the predisposing factors of the earthquake-induced landslides would be definitely of interest but it is far beyond the scope of this work.

Minor specific comments and technical corrections: we clarified that the debris flows were transformed from landslides with the help of antecedent, we added the information on the grain size distribution to Figure 6, we included the effect of partial saturation to Figure 12 and we corrected the spelling of "debris flows".

Please find the manuscript enclosed with track changes.

Best regards, Velio Coviello

Please also note the supplement to this comment:
https://esurf.copernicus.org/preprints/esurf-2020-36/esurf-2020-36-AC3-supplement.pdf

**Supplement:**

[revised manuscript text omitted]

---

## Author Response (AR2)

Point-to-point reply to reviewer (R2)

Manuscript esurf-2020-36 "Earthquake-induced debris flows at Popocatépetl Volcano, Mexico"

Overview and general comments:

Manuscript studies the co-seismic landslides that are triggered by the Puebla-Morelos EQ (MW 7.1). The study suggests that the volcanic landscape could further enhance the impact of the EQ. It is a comprehensive report of the event, however, the results seem to lack novelty. The method of the work is not clearly described, it should be extended beyond postprocessing the used data. Additionally, the text is hard to follow there are several redundant information and no clear line that leads to the take home message of the manuscript—"new insights to constrain a multi-hazard risk assessment". The study fails to show, how this result is reached. Hence in its current form I am afraid that the manuscript won't attract its deserved attention by the community.

Major Issues:

These landslides are some ~70km away from the epicenter, which is quite large distance to trigger landslides. Are they somewhat closer to the rupture fault?

*We thank the reviewer for this comment that gave us the input to clarify the relevance of the mass wasting that occurred on Popocatépetl Volcano in 2017. In general, triggering of landslides is more a matter of intensity of ground motion and not of the simple distance from the epicenter or rupture fault. Measured and interpolated data show that seismic shaking at Popocatepetl was particularly strong, with PGA of 106.83 cm/s^2 (Fig. 1c and 2). Thus, should not be surprising that the seismic event triggered landslides and generated surface ruptures on volcano slopes. In addition, the ground motion during the 2017 earthquake was anomalously large in the frequency range 0.4–1 Hz as intraslab earthquakes involve higher stress drop than their interplate counterparts (Singh et al., 2018). Consequently, the ground motion was relatively enriched at high frequencies as compared with that during interplate earthquakes, which is dominated by lower frequency waves (f< 0:5Hz), and this can contribute explaining the high value of measured PGA on the volcano slope. We added this information in the sections 3 The MW 7.1 intraslab Puebla-Morelos earthquake and 5.2 Initiation of co-seismic landslides.*

Reviewer 2 suggested authors to reflect from other global earthquake triggered landslides, though Hokkaido EQ might not be the best example, since the case study in the manuscript is a volcanic region, hence I recommend authors to rather take a look in such cases, e.g. Kumamoto 2016 EQ. von Specht et al. (2019) studies co-seismic landslides around the Aso Caldera, they suggest a velocity-based model to study the co-seismic landslides; while Fan et al. (2019; 2020) are the most up to date comprehensive review about the recent developments in the area of exploring co-seismic landslides. These articles could help authors improving their work in introduction and in discussion.

Fan, X., Scaringi, G., Korup, O., West, A. J., Westen, C. J., Tanyas, H., et al. (2019). Earthquake-Induced Chains of Geologic Hazards: Patterns, Mechanisms, and Impacts. Reviews of Geophysics, 57(2), 421–503. https://doi.org/10.1029/2018RG000626

Fan, X., Yunus, A. P., Scaringi, G., Catani, F., Subramanian, S. S., Xu, Q., & Huang, R. (2020). Rapidly evolving controls of landslides after a strong earthquake and implications for hazard assessments. Geophysical Research Letters. https://doi.org/10.1029/2020GL090509

von Specht, S., Ozturk, U., Veh, G., Cotton, F., & Korup, O. (2019). Effects of finite source rupture on landslide triggering: the 2016 MW 7.1 Kumamoto earthquake. Solid Earth, 10(2), 463–486. https://doi.org/10.5194/se-10-463-2019

*We agree. The revised version of our manuscript presents an improved introduction and discussion, with reference to other case studies of earthquake triggered landslides, including your recent articles (Fan et al., 2019, 2020) and other papers (Kumamoto, 2016; Specht et al., 2019; LaHusen et al., 2020; Schulz et al., 2012; Serey et al., 2019; Wartman et al., 2013). Also, we tried to better present the peculiarity of the MW 7.1 Puebla-Morelos intraslab earthquake and its impact on landsliding. The impact of subduction-related earthquakes is known to be smaller than events triggered by shallow crustal earthquakes but in this paper we show how an unusual intraslab earthquake can produce an exceptional impact on an active volcano.*

Abstract:

Reviewer 1 suggested rewriting the abstract and clearly describing the findings and interesting point. Although I fully agree on the 1st reviewer, I see inadequate minor changes in the abstract with nearly no information on the findings of the current work. It lacks any clear result of the conducted work; it just reports that the observations are unique and the results call for reassessing the multi-hazard risk assessment concept in the region. The abstract does not lead to this suggestion, authors should explain how it is reached. In the meantime, the abstract is somewhat long, and written in a mixed manner: it follows, study site information and event description, research questions, again study site information and event description, method, and nearly no results. Additionally, I recommend authors to use active voice instead of passive sentences in the abstract, also throughout the manuscript.

*We reworded the abstract. Now, after some brief information about MW 7.1 Puebla-Morelos intraslab earthquake, we clarified that: (i) we produced a landslide map based on a semi-automatic classification of a 50-cm resolution optical image, (ii) we identified hundreds of shallow soil slips and three large debris flows for a total affected area of 3.8 m^2, (iii) we identified a transient reactivation of local faults and extensional fractures as one of the mechanisms that has weakened the volcanic edifice and promoted the largest slope failures, and (iv) we reconstructed this mass wasting cascade by means of original field evidences, samples from both landslide scarps and deposits and analysing of remotely sensed and rainfall data. All of these are new, previously unpublished results.*

Introduction:

I recommend authors to improve their introduction fundamentally, in its current state, it is weak in content and involves a lot of interpretation about the study area. Authors mix several perspectives about volcanic landslides and EQ-induced landslides, which makes it hard to follow. Authors frequently use serious of citations to support their arguments, which indeed do not prepare the reader to the content of the manuscript. For example the sentence "In the following years, a growing number of studies started focusing on the impact of landsliding caused by large-magnitude earthquakes on the sediment yield (e.g., Pearce and Watson, 1986; Dadson et al., 2004; Marc et al., 2019).". What is important on these articles for the purpose of the current study? There are a few more similar examples in the introduction.

*We completely reorganized the introduction following a more clear structure. After some lines dedicated to earthquake-induced landslides, with particular attention given to the role of fault rupture mechanism, we address the topic of slope instabilities of active volcanoes. Finally, we introduce the mass wasting episode triggered by the 2017 MW 7.1 Puebla-Morelos intraslab earthquake at*

*Popocatépetl volcano. We splitted the test describing the Popocatépetl Volcano and the 2017 Puebla-Morelos earthquakes in two dedicated sub-sections belonging to the new section 2 Background.*

Data and methods:

This section sound like a data and post processing of that data, there is no method in it. Authors use frequently passive voice in the section, if the NDVI, the timing of the landslides, and radiometric calibration are adapted from another work, they should be cited. Field data sounds to be collected by someone else as well, authors should cite those sources, if they did it themselves, then better to use active voice. A dedicated method section is necessary.

*We reorganized this section. First, we declare that we adopted a combined field- and remote-based approach to retrieve timely and original information about the earthquake impact on such a harsh environment. Then, we improved the figure presenting the landslide map adding in the background the map of the PGA distribution, we added a new figure with details of the three areas of the Popocatépetl volcano more affected by coseismic landslides. We use now the active voice in the text, thanks for this suggestion, to clarify that all the information presented within this section was collected and elaborated by the authors of the manuscript.*

Timing of the events:

Authors frequently refer to some locations, e.g. Amecameca and Atlautla, it would ease following the text if there would be a general introductory image.

*Done, we added this information in Fig. 1.*

Authors should consider using the Soil Water Index to estimate the soil wetness, a option is to use the method suggested by Chen et al. (2017). Chen, C.-W., Saito, H., & Oguchi, T. (2017). Analyzing rainfall-induced mass movements in Taiwan using the soil water index. Landslides, 14(3), 1031–1041. https://doi.org/10.1007/s10346-016-0788-1

*Thanks for the suggestion but we feel that using data from a single raingauge located at a distance > 10 km from the source areas of the landslides to calculate the Soil Water Index would result in too large uncertainties and do not add a quantitative value to the analysis.*

Initiation of co-seismic landslides:

Authors refers frequently to seismic amplification justifiably, why it is not shown in the manuscript? In the study area it might not be possible to compute it precisely due to lack of data, in that case authors could at least show the total curvature, which is the base of the site amplification.

*As stated in Sections 2 and discussed in section 5.2, the PGA at PPIG station was very high (106.83 cm/s^2), and nearly two times more than the one recorded at station CU (57.1 cm/s^2). Thus, the value of PGA at PPIG station cannot be justified without local effect considering the distance from the epicenter. This supports the hypothesis that the amplification of the seismic shaking occurred on the volcano flanks due a combination of local (i.e., due to slope material properties) and topographic effects. We have the impression that providing a value of amplification computed based on the total curvature would not add a relevant supporting argument as the complex topography of the volcano*

*flaks likely plays a major role in terms of local ground motion. We argue that the PGA at the source areas of the major landslides was higher than the measured PGA at PPIG due to the combined effect of topography and of the shorter distance from the source. This is consistent with the map of PGA distribution produced by USGS (see new Figure 2).*

Line 322—323: von Specht et al. (2019) shows impact of the site amplification on landslide location on the landscape.
von Specht, S., Ozturk, U., Veh, G., Cotton, F., & Korup, O. (2019). Effects of finite source rupture on landslide triggering: the 2016 MW 7.1 Kumamoto earthquake. Solid Earth, 10(2), 463–486. https://doi.org/10.5194/se-10-463-2019

*Done, we added this and other references (e.g., Del Gaudio and Wasowski, 2007) to the list.*

Minor suggestions:

Abstract:

In the abstract magnitude of the Puebla-Morelos EQ is specified only with "M" initial, isn't it commonly referred as "MW"?

*Done ($M_W$) throughout the manuscript.*

Introduction:

Line 29: the first sentence of the Introduction does not really lead anywhere, EQs and rainfall are certainly triggering landslides independently. I assume authors want to emphasize that antecedent rainfall could enhance the total co-seismic landslides. Or do they mean subsequent rainfall could further trigger landslides on those EQ weakened hillslopes?

*The first sentence was changed to avoid misunderstanding. What was meant is that both processes (earthquake and rainfall) can trigger landslides independently, but now in the introduction we start focusing on the role played by earthquakes, thus the second part of the sentence has been deleted (… and rainfall events …).*

Line 55–67: involves some interpretation about the study area. It is better to move it to the dedicated section.

*As stated before, we created a new dedicated section (2.1 Popocatepetl volcano) to describe the study area where we moved these and other lines.*

Line 60: "volcano's summit" --> wrong use of possessive apostrophe. I have witnessed some others throughout the manuscript, please pay attention to its correct use.

*Done, we went carefully through the manuscript to correct other typos.*

Context and study site:

A better title would be "Study area" only. It would be nice if the authors could edit and shortened the section a bit, but it is not necessary.

*Answer: See previous answers about the new section 2.1 Popocatepetl volcano.*

Landslide mapping:

C14 dating is first mentioned in the results section, I guess it should be listed also in a dedicated method section.

*Done, we added these sentences to the section 3 Data and methods: "We selected a soil sample for radiocarbon analysis to identify the age of the stratigraphic sequence and to define its distribution. The 14C age has been obtained through accelerator mass spectrometry dating (BETA Analytic Laboratory) and calibrated with the IntCal20 calibration curve (Reimer et al., 2020)".*

Line 163: How many landslides, their total area or volume? Please cite the paper/work, who conducted the analyses?

*We now provide the total area of earthquake-induced landslides at Popocatepetl (3.8 km^2) and we clarified that we did this analysis (unpublished results).*

Line 168–171: The sentence is hard to understand, please paraphrase.

*Done: "Pumice-fall deposits consist of open-framework, clast-supported units, composed of gravel-sand sized fragments of pumice embedded in a matrix of fine material (from silt to clay)".*

Line 181–183: The sentence is hard to understand, please paraphrase.

*Done: "Here, a main unit of pumice-fall deposit (C in Figure 6d) features a total thickness of 3.5 m, and consists of a massive, clast-supported unit dominated by coarse fragments barren of any silt and clay fractions. This latter unit lies on a 10 cm-thick sandy layer (B in Figure 6d)".*

Line 184: Cite the source paper of this C14 dating.

*Done, we added Reimer et al. (2020) to the reference list.*

Characterization of debris flows and associated deposits:

Line 205–206: are these the 5 landslides that were mentioned at the beginning of the previous section?

*Yes, they are now mentioned in the revised version of the manuscript.*

Discussion:

This section should also highlight the contribution of the authors to the existing literature. It currently focusses only on understanding the reasons of landsliding on Popocatépetl Volcano.

*We expanded the discussion, especially sections 5.2 and 5.3:*

*"The ground motion during the 2017 earthquake was anomalously large in the frequency range 0.4–1 Hz as intraslab earthquakes involve higher stress drop than their interplate counterparts (Singh et al., 2018). Consequently, the ground motion is relatively enriched at high frequencies as compared with that during interplate earthquakes, which is dominated by lower frequency waves (f< 0:5Hz), and this can contribute explaining the high value of PGA measured on the volcano slope".*

*"during the earthquake the headwaters of Hueyatlaco, Huitzilac, and Xalipilcayatl ravines could have experienced even higher values of PGA due to the effect of topographic amplification of seismic waves. The PGA map produced by the USGS Seismic Hazard Program shows values between 0.28g in the southern sector of the cone and up to 0.18g closer to the vent (Figure 2c). The spatial interpolation of PGA clearly shows the interaction between the energy distribution and the topography, which played an important role in the location of landslides. The cluster of smaller landslides located on the southwestern side of the volcanic cone, closer to the epicenter, is likely due to the combination of large ground motion (Figure 2) and high slope (Figure 3b)".*

*"Finally, it is worth mentioning that during our last field campaign of November 2019 we observed that the source areas of the larger debris flow that occurred at Hutzilac ravine were becoming stable thanks to the combined effect of the removal of fine material and of the growth of new vegetation. On the contrary, the large amount of material deposited along the channel remains available for remobilization for many years, resulting in a remarkable increase of sediment yield as observed in other locations (e.g., Fan et al., 2021). Indeed, during 2018 and 2019 rainy seasons, the fine sediment remobilized from the debris-flow deposits in Huitzilac ravine reached the road connecting San Juan Tehuixtitlán to Atlautla".*

Line 312–315: some supporting references would cement these nice arguments.

*Done, we reworded as follows "Topographic amplification of ground vibrations is due to the reflection/diffraction of seismic waves, which are progressively focused upwards and the constructive interference of their reflections and the associated diffractions increases towards the ridge crest, giving rise to enhanced ground accelerations on topographic highs (Bouchon et al., 1996; Davis and West, 1973; Meunier et al., 2008)".*

Figures:
Supporting text, e.g. axis ticks, are very small to read nearly in all the figures.

Figure 1: Would it be possible to show also the ruptured fault?

*We added the focal mechanism as a marker of the epicenter location and a legend defining the main markers that we used.*

Figure 2: Please mark subplots by a, b, c, d.

*Done to all figures.*

Figure 3: Landslides are rather tiny in the image, 1 or 2 more subplots zooming in them could help.

*We added a new figure (Figure 3) with details of the areas of the volcano more affected by landsliding.*

Figure 4: Please highlight the PO1906, 1927, and 1911 in subplot a. would it be possible to show the

outline of the landslides in subplot b, or show additionally Overhead view From Google Earth or satellite images.

*We improved this figure (all sampling points renamed and labeled in panel a) we added in panel c an aerial view of the two scarps profile reported visible in panel b.*

Figure 6: Subplot b and d: the marks of A, B, C, and D are not contrasting enough to spot them easily. Subplot e refers to the figure 4a, I assume?

*We increased the size and changed the color of labels in panels b, c, d and we added a sketch representing the grain size distribution of each stratigraphic section (panel e).*

Figure 7: Really informative image, helps reader to follow nicely, please mark the subplots with a, b, c... and link them to table 1 in the figure. Another option could be to integrate table 1 inside this figure, it is of course the choice of the authors.

*Done, we added a, b, c... to the figure.*

Figure 8: Please try to link the figure to figure 4a

*We added a label with the sample code and a grain distribution plot of the deposits in the three ravines.*

Figure 10: what do the different colors of the bars mean?

*We improved the figure, we use different blue shades to highlight the different months, the red bar (rainfall accumulated on 17 September) is defined in the caption.*

Tables:
Table 1: What is the slope unit?

*We added the unit (degree).*

Table 2: Involves a lot of numerical information, it is hard to understand, would it be possible to convert it to a figure, e.g. bars?

*We moved the information on grain size distribution of landslide scars to Figure 6e and of landslide deposits to Figure 8i, the complete numerical information is provided in Appendix A.*

Table 3: It could be merged with table 1, or better both of them in Figure 7.

*We prefer to keep these tables separated as they refer to landslide scarps and debris flow deposits respectively, and they are presented in two different sections.*